# Interpreting nanovoids in atom probe tomography data for accurate local compositional measurements

Xing Wang[1✉], Constantinos Hatzoglou[2], Brian Sneed[1], Zhe Fan[3], Wei Guo [1], Ke Jin[3], Di Chen[4], Hongbin Bei[3], Yongqiang Wang [4], William J. Weber [3,5], Yanwen Zhang [3,5], Baptiste Gault [6,7], Karren L. More [1], Francois Vurpillot [2] & Jonathan D. Poplawsky[1✉]

Quantifying chemical compositions around nanovoids is a fundamental task for research and development of various materials. Atom probe tomography (APT) and scanning transmission electron microscopy (STEM) are currently the most suitable tools because of their ability to probe materials at the nanoscale. Both techniques have limitations, particularly APT, because of insufficient understanding of void imaging. Here, we employ a correlative APT and STEM approach to investigate the APT imaging process and reveal that voids can lead to either an increase or a decrease in local atomic densities in the APT reconstruction. Simulated APT experiments demonstrate the local density variations near voids are controlled by the unique ring structures as voids open and the different evaporation fields of the surrounding atoms. We provide a general approach for quantifying chemical segregations near voids within an APT dataset, in which the composition can be directly determined with a higher accuracy than STEM-based techniques.

[1] Center for Nanophase Materials Sciences, Oak Ridge National Laboratory, Oak Ridge, TN, USA. [2] Normandie Université, UNIROUEN, INSA Rouen, CNRS, Groupe de Physique des Matériaux, 76000 Rouen, France. [3] Materials Science and Technology Division, Oak Ridge National Laboratory, Oak Ridge, TN, USA. [4] Materials Science and Technology Division, Los Alamos National Laboratory, Los Alamos, NM, USA. [5] Department of Materials Science and Engineering, University of Tennessee-Knoxville, Knoxville, TN, USA. [6] Max-Planck-Institut für Eisenforschung, Max-Planck-Str, 1, 40237 Düsseldorf, Germany. [7] Department of Materials, Imperial College London, Royal School of Mine, London SW7 2AZ, UK. ✉email: xvw5285@psu.edu; poplawskyjd@ornl.gov

Nanosized voids, including pores, cavities, and bubbles, are common defects found in materials[1]. Voids can be generated under a variety of service conditions and are most often detrimental to the material's structural properties. For example, a high strain rate leads to the nucleation and growth of nanovoids, which can coalesce and cause the fracture of ductile metals[2,3]. Helium (He) bubbles are formed in materials used in nuclear reactors owing to the synergistic effects of neutron bombardment and the nuclear (n,α) transmutation reactions, resulting in swelling of the material[4,5]. Besides such undesirable voids, nanosized voids can be intentionally introduced to optimize a material's properties[6]. For example, nanostructured porous silicon has found extensive application in optoelectronics and biomedicine owing to its high-efficiency photoluminescence and variable surface chemistries[7]. The inherent pore structures in zeolites are exploited for catalysis owing to their high surface-to-volume ratio[8].

Voids are essentially empty volumes and their interactions with the surrounding matrix are controlled by the surrounding atoms. An accurate characterization of the material chemistry around voids is important for both mitigating the formation of unintended voids and taking advantage of the presence of intended voids. Scanning transmission electron microscopy (STEM)-based spectroscopy methods, e.g., electron-energy loss spectroscopy (EELS) and energy dispersive X-ray spectroscopy (EDS), are powerful tools for performing compositional analysis at the Å-level[9,10]; however, when the size of a target object is smaller than the specimen thickness, the obtained composition of the target is an average value that includes all of the material within the full sample thickness, not just the target because the electron beam is transmitted through the entire sample thickness, leading to errors in the compositional measurement of the targeted feature. Atom probe tomography (APT) provides sub-nm-scale three-dimensional (3D) chemical information with a much higher compositional sensitivity than that measured by analytical STEM techniques[11]. APT analyses are enabled by applying a large electrical field to a needle-shaped specimen tip that ionizes and pulls the atoms off the tip's surface one-by-one, i.e., field evaporation[11]. The type and position of the ionized atoms are determined using time-of-flight mass spectrometry, a position-sensitive detector, and a reconstruction model[12]. This makes APT a promising approach for determining local compositions without the detrimental influence of sample thickness. APT has been applied to characterize the material chemistry near nanoscale defect features such as grain boundaries and precipitates[13–16]. Further details regarding the APT experiment can be found in the Methods section.

During an APT experiment, the specimen must have a perfectly hemispherical cap shape for the magnification to be consistent on the detector. Any deviation from a hemispherical surface causes the projected surface image to be distorted (aberrated). Aberrations are commonly encountered in APT reconstructions near structural or chemical heterogeneities that may change the normal field evaporative behaviors of atoms. These aberrations are likely exacerbated for voids because the cavity structure can abruptly deform the APT tip morphology. Overall, APT data analysis is challenging for materials that cause the APT tip surface to deviate from a hemisphere during the experiment and is particularly challenging for void-containing materials.

A few studies have employed APT to analyze voids in materials; unfortunately, the results suggest that producing accurate reconstructions near these cavity structures and interpreting the 3D data are challenging[17–21]. Larson et al.[17] indicated that the local magnification effect near a void can introduce large aberrations in the atom positions that broaden the void using field-evaporation simulations. Edmondson et al.[19] studied nanosized bubbles in ferritic alloys using APT and found that the bubbles could be visualized using iso-density surfaces at approximately half the atomic density of the alloy, indicating bubbles were shown as low-density regions. Porous zeolite catalysts were recently investigated using APT by two independent research teams[22–25]. Although pores have zero atomic density inside, i.e., empty space, no such regions were found in the APT reconstructions[23]. Rather, spatial correlations of regions with high densities and high concentrations of segregated elements in the proximity of the pores suggest that pores may present as high-density regions in the 3D reconstruction. Thus, it remains unclear whether voids will appear as low-density or high-density regions in APT reconstructions. To obtain accurate compositional information around nanovoids using APT, it is necessary to more thoroughly understand the field-evaporation process near voids and its effects on the atom coordinates during APT data reconstruction.

To this end, we conducted correlative APT-STEM experiments that were combined with simulated APT experiments for a series of void-containing alloys. Because visualizing voids in an APT reconstruction is a concern, a needle-shaped APT tip was imaged by STEM prior to conducting the APT experiment. The size and location of each void within the needle-shaped specimen were accurately determined from STEM images acquired at different rotation angles. The same tip was then field-evaporated and analyzed using APT such that the local atomic density variations shown in the APT reconstruction could be directly correlated with the voids observed in the STEM images. A simulated APT data set reproduced and explained the observed atomic density variations by using experimentally measured void sizes and compositional segregation as input parameters. Based on the insights obtained from the experiments and simulations, we provide approaches for interpreting compositional and dimensional information of nanovoids using APT, in which the compositional measurements can achieve a much higher accuracy than conventional STEM-based EDS and EELS measurements.

## Results

**Correlating local atomic density variations in APT to voids.** The materials studied are Ni and a series of Ni-based single-phase-concentrated solid solution alloys (SP-CSAs) that include NiFe, NiCoCr, and Si- and Al-doped NiCoCrFe. Overall, SP-CSAs possess an exceptional combination of high strength, ductility, and resistance to radiation damage, making them promising candidates for next-generation nuclear reactors[26,27]. In addition, obvious compositional segregations associated with voids exist in these SP-CSAs[28–30], creating a segregation shell, which can be used to mark the void position and help determine the flight paths of shell atoms during the APT experiment. Ion irradiation of the SP-CSAs using He or Ni ions was performed to generate nanosized He bubbles or voids. Helium bubbles are simply cavities with pressurized He inside[30]. As the bubbles rupture during an APT experiment, the He atoms within the bubble are immediately released into the vacuum chamber, and bubbles will undergo the same evaporation process as voids in APT. For simplicity, He bubbles will be called voids in the following text unless otherwise specified. Preliminary segregation measurements near the voids were performed using STEM-EDS. The general segregation trend is that Cr and Fe deplete around the voids. This trend agrees well with predictions based on density functional theory calculations, which suggest that vacancy migration barriers via Cr and Fe lattice sites are lower than Ni and Fe sites in the SP-CSAs, i.e., vacancies prefer to diffuse via Cr or Fe atoms[31]. During void growth, the influx of vacancies towards the void surfaces and the outflux of Cr and Fe atoms leads to the observed segregation. An example of a STEM-EDS map near a void in NiCoCr is

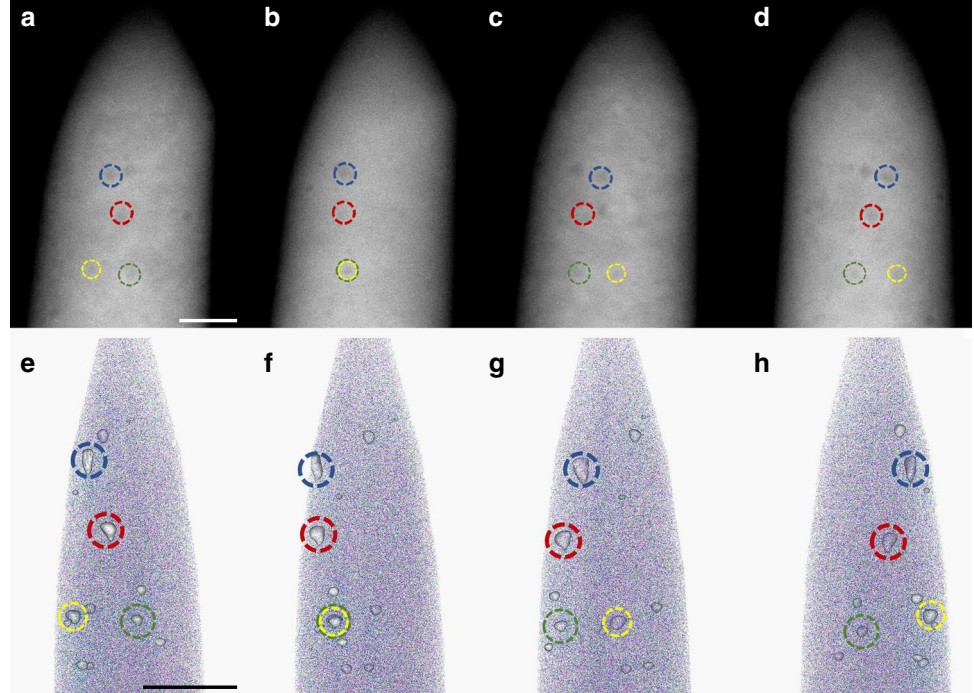

**Fig. 1 HAADF-STEM and APT images acquired for a NiCoCr needle-shaped specimen at different rotation angles.** STEM and APT images are shown on the top and bottom panels of the fig., respectively. **a** and **e** are at 0°; **b** and **f** are at 60°; **c** and **g** are at 120°; **d** and **h** are at 180°. In the APT reconstructions, 50 atoms/nm³ iso-density surfaces are shown in white. The same voids are marked by circles of the same color in both the HAADF-STEM image and APT reconstruction. The scale bar for both rows is 40 nm. Note the APT reconstruction only represents the central part of the tip in the HAADF image because only ~60% of the needle is captured in an APT experiment and a focused-ion beam cleaning was performed after the STEM (see Methods for details).

provided in Supplementary Fig. 1, which shows that the segregation is maximized right at the void location with 26 at.% Cr and 38 at.% Co (Ni is ~36 at.%, but is not segregated near the void). Note that these EDS-measured concentrations are average values from both the segregated shell around the void and the homogeneous matrix above and below the void, so the real composition of the segregation must be higher than the EDS results.

For the correlated APT-STEM study, the tip of a needle-shaped APT specimen was imaged by acquiring a tilt series in the STEM. A high-angle annular dark-field (HAADF) image was acquired at 5° tilt increments over a 180° tilt range. Figure 1a–d show four typical HAADF-STEM images from the NiCoCr needle acquired at rotational angles of 0°, 60°, 120°, and 180°. Voids appear as dark circles in the HAADF-STEM images because there are either no atoms or only light atoms (He in bubbles) within the cavity structures. The complete rotation series comprised of 37 HAADF-STEM images is presented in Supplementary Movie 1. Based on this image series, we are able to track the position of each void as the needle-shaped specimen rotates. Four voids are marked by colored circles in the HAADF-STEM images in Fig. 1a–d.

After the STEM measurement, the same needle-shaped specimen was transferred into a local electrode atom probe (LEAP) to perform the APT experiment. Similar to results in the literature[19,32], empty space cannot be unambiguously identified in APT reconstructions of the void-containing specimen. Interestingly, there are clear density variations within the APT data, which can be highlighted using iso-density surfaces as shown in Fig. 1e–h, where 50 atoms/nm³ iso-density surfaces are used to highlight the high-density regions (the average density for the APT reconstruction is 30 atoms/nm³). These high-density regions consistently appear cone-shaped with the vertex pointing toward the bottom of the needle, and the position of each cone matches

well with the positions of the voids shown in the HAADF-STEM images. The APT reconstructions in Fig. 1e–h are shown at the same rotational angles as the HAADF-STEM images. For easy comparison, several high-density regions are marked by circles of the same color in both sets of images, clearly showing that the high-density regions observed by APT correspond to the same voids observed by STEM imaging. An animation is provided in Supplementary Movie 2 to compare the complete rotational series of 37 HAADF-STEM images and the corresponding APT reconstruction with 50 atoms/nm³ iso-density surfaces.

The correlative APT-STEM experiments clearly show that voids do not present as empty space in these APT reconstructions. Rather, the local atomic density shown in the APT reconstruction increases at the void position compared with the alloy matrix. More interestingly, APT analyses of other void-containing materials reveal that the local atomic density variations at a void exhibit two characteristic patterns. To show these two patterns, cylindrical regions-of-interest (ROI) that pass through the center of voids in the APT reconstruction are used to produce the 1D density profiles shown in Fig. 2a, b. In Fig. 2, the $z$ axis is parallel to the evaporation direction, so a region with a lower $z$ value means the region is closer to the top of the specimen and evaporates earlier during the APT experiment. For ease of comparisons between different samples, in Fig. 2 we plot the reduced density, which is the local atomic density divided by the average atomic density in the matrix. For voids in NiCoCr, NiFe, and Ni, the local density rapidly increases and then slowly decreases to the matrix value (Fig. 2a). This type of density variation is designated as λ type because of the shape of the 1D density profile. Figure 2a is from NiCoCr; examples for Ni and NiFe are provided in Supplementary Fig. 2. Voids in doped NiCoCrFe exhibit a different local density variation, where the local density is lower than the matrix but has a high-density peak in the center of the void (Fig. 2b). This type of variation is designated as ω type

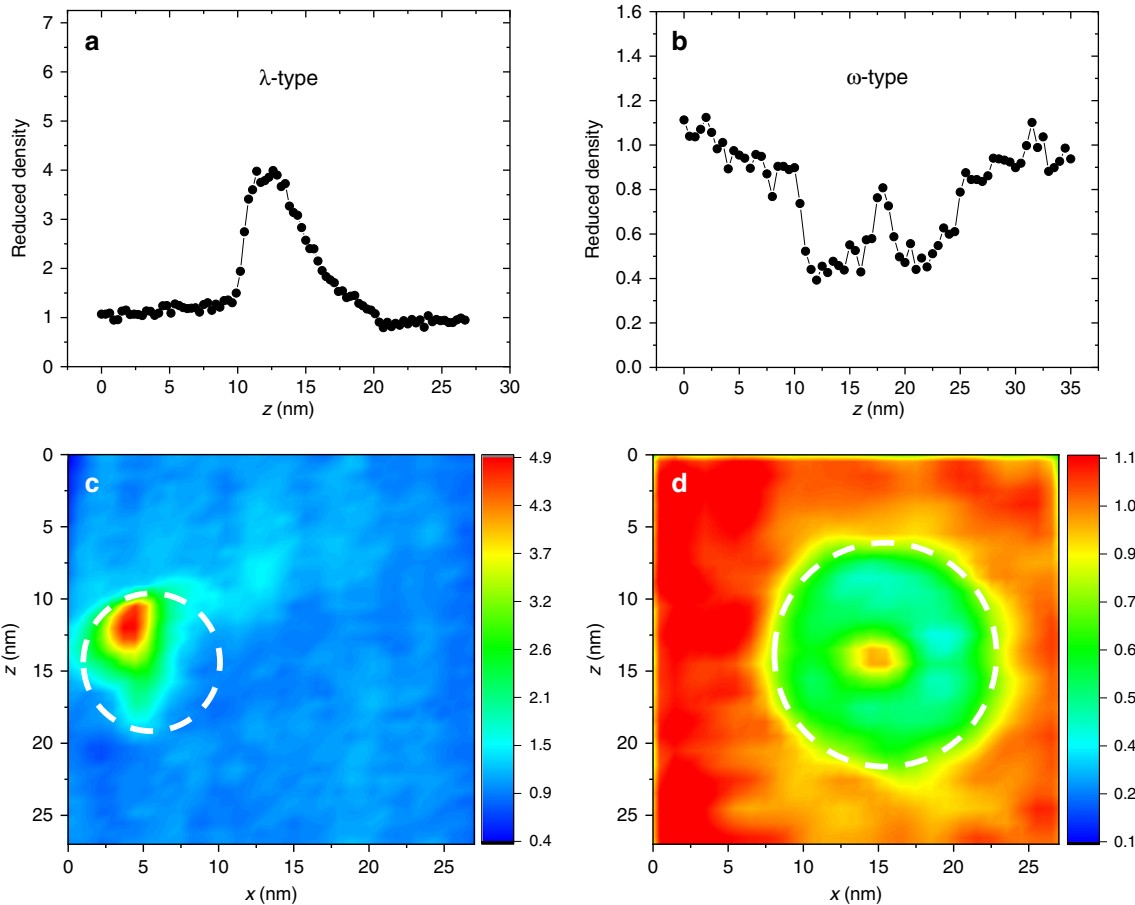

**Fig. 2 1D local density profiles and 2D density contour plots at a void location. a** and **c** are results from NiCoCr; **b** and **d** are results from doped NiCoCrFe. To produce 1D profiles, 5 nm diameter cylindrical ROIs are oriented perpendicular to the hemispherical surface of the APT reconstruction and pass through the void. 27 × 27 × 3 nm cuboidal ROIs through the center of the voids are used to calculate the 2D density contour plot.

because of the shape of the 1D density profile. To fully visualize differences in density variations, 2D density contour plots are calculated using a cuboidal ROI that sliced through the center of the voids (Fig. 2c, d). The white dashed circles represent the void location determined from the corresponding HAADF-STEM images and/or the elemental segregation profiles. For the λ-type density variation (Fig. 2c), the high-density region presents as an inverted triangular shape near the top of the void; for the ω-type variation (Fig. 2d), the 2D density contour map exhibits a circular shape with a higher density core and a lower density shell in the outer layer of the void. The observed density variations are independent of the APT experimental running conditions, which has been verified by the experimental results acquired in both laser and voltage modes (Supplementary Fig. 3). The different types of density variations introduced by similar voids formed in different materials indicate that both the cavity structure and the material chemistry around the void play an important role in dictating the field-evaporation process near voids.

**Voids in a simulated APT experiment**. It is counterintuitive that a void, which is essentially empty space, can exhibit an increased local atomic density region at the void position in the APT reconstruction. Simulated APT experiments provide a unique approach to gain insight on how the field-evaporation process is affected by the void's structure and surrounding composition[33]. Such simulations reproduce the gradual evolution of a virtual APT specimen by evaporating the tip atom-by-atom with an evaporation field assigned to each element comprising the

material. The electric field over the evolving specimen surface is dynamically computed assuming a constant voltage applied to the tip. The distribution of the electric field is used to calculate the trajectories of the ionized atoms from the specimen tip to the APT detector. The detector impact position and elemental nature are used to reconstruct the 3D structure of the virtual APT specimen in a similar way as for an experimental dataset[34]. Here, the field-evaporation simulation model developed by Vurpillot et al.[34] is used in our study. NiCoCr is chosen as the model alloy and nanosized voids with segregated shells are embedded within a virtual APT specimen to mimic our experimental observations. Simulation details are provided in the Methods section, and the simulated tip morphology with the electric field distribution on the surface are displayed in Supplementary Fig. 4.

The evaporation field is an important input parameter for the field-evaporation simulations. Because the segregated shell surrounding the void has a different composition from the alloy matrix, and these atoms are exposed to the internal surfaces of the void, it is reasonable to expect that the atoms forming the shell will have a different evaporation field from the matrix atoms. We took the possible differences into account by performing three simulations where the evaporation field of the shell was (1) 30% higher than the matrix, (2) 30% lower than the matrix, and (3) equal to the matrix. The atom distributions in the three simulated APT reconstructions near the nanosized void are shown in Fig. 3a–c. In all three simulations, the void, which should be empty, is filled with atoms from the matrix similar to the observation based on the correlative STEM-APT experiments. In addition, detailed features

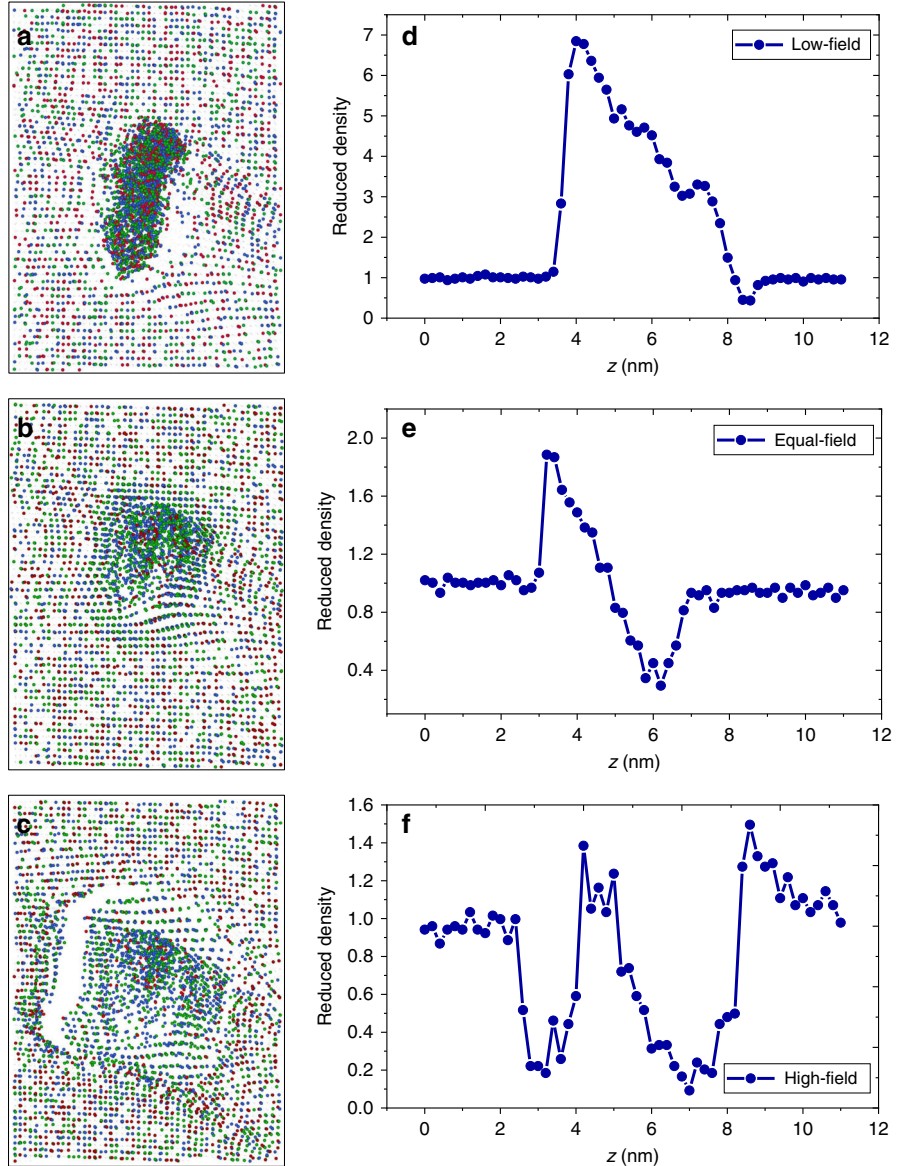

**Fig. 3 Atom distributions and 1D local density profiles near voids based on field-evaporation simulations. a** and **d** show results with $E_{shell} = 0.7\, E_{matrix}$; **b** and **e** show results with $E_{shell} = E_{matrix}$; **c** and **f** show results with $E_{shell} = 1.3\, E_{matrix}$. Atom distributions in **a**, **b**, **c** are from 11.2 × 8.3 × 1 nm cuboidal ROIs near the void projected onto the $x$ direction in the simulated APT reconstructions. Each dot represents one atom, with Co, Ni, and Cr atoms shown as blue, green, and red dots, respectively. 1D profiles are calculated using a 11 nm long × 3 nm diameter cylindrical ROI through the void center and are oriented parallel to the evaporation direction of the simulated APT reconstruction.

of the atom distributions are clearly different depending on the shell evaporation field. In the low-evaporation-field case, a high-density region with an inverted triangular shape forms near the top of the void (Fig. 3a), which is almost identical to the experimental 2D density map shown in Fig. 2c; in the high-evaporation-field case, the atom distribution exhibits a high-density core and low-density shell inside the void (Fig. 3c), which is similar to the experimental 2D density map shown in Fig. 2d. Similarities between simulation and experiment are also observed in the 1D density profiles. The calculated 1D density profiles from the simulated APT reconstructions exhibit a λ shape in the low-evaporation-field case (Fig. 3d) and an ω shape in the high-evaporation-field case (Fig. 3f), which matches the experimental profile shapes exactly in Fig. 2a, b. In the equal-evaporation-field case, the atom distribution (Fig. 3b) and 1D density profile (Fig. 3e) show features that are similar to the other two cases. For example, a high-density region is evident as the void opens during field evaporation (low-field case) and a low-density

region is evident as the void is field-evaporated (high-field case). Although an experimental counterpart for the equal-field case was not found, the good match in the characteristic density variation shapes between experiment and simulation for the high- and low-field cases demonstrates that the evaporation fields of shell atoms control the local density variation patterns in the APT reconstruction. Thus, for materials that exhibit a λ-shape variation, including Ni, NiFe, and NiCoCr, it is expected that the shell atoms have a lower evaporation field than the matrix, while for doped NiCoCrFe, which shows a ω-shape variation, the shell atoms have a higher evaporation field. A more-detailed analysis showing a series of local 1D density profiles with incremental shell evaporation-field steps is provided in Supplementary Fig. 5.

**Charge state ratios of voids in APT data.** Experimental evidence was found to support the expected relation between shell

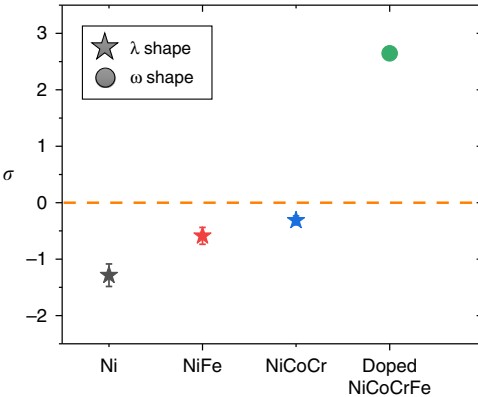

**Fig. 4 Logarithmic ratio of $Ni^{2+}/Ni^{1+}$ between the matrix and void shell for different alloys studied.** Error bars for each data points are from standard deviation of measured ratios from multiple voids in each sample.

evaporation field and local density variation. According to Kingham's post-ionization theory, an ion emitted from a material surface owing to the applied electric field can be further ionized by tunneling of electrons from the ion into the material surface[35]. As the applied field strength increases, the probability of post-ionization increases and the average ion charge state rises[36]. In an APT specimen containing regions having different evaporation fields, atoms ionized from a higher evaporation-field region will show a higher charge state. In our APT experiment, most of the collected ions are in either 1+ or 2+ charge state; thus, we use the Ni ions as an indicator and calculate the ion number ratio of $Ni^{2+}$ to $Ni^{1+}$ from spherical ROIs both at the void position and in the matrix. To compare with different materials, a logarithmic ratio of the void shell to the matrix charge state ratio is defined as:

$$\sigma = \ln \frac{(Ni^{2+}/Ni^{1+})_{void}}{(Ni^{2+}/Ni^{1+})_{matrix}}. \qquad (1)$$

If $\sigma$ is below zero, the shell atoms around the void have a lower evaporation field than the matrix, and vice versa. The obtained $\sigma$ for different alloys are summarized in Fig. 4. For alloys that exhibit a λ-shape variation, including Ni, NiFe, and NiCoCr, $\sigma$ values are all below zero, whereas in Al- and Si-doped NiCoCrFe, where a ω-shape density variation is observed, $\sigma$ is above zero and agrees with the expectation that the evaporation field near the void is higher than that for the matrix. The reason for the different evaporation fields in different samples is still under investigation, and possible explanations are provided in Supplementary Fig. 6. Nevertheless, the current analyses demonstrate that the relative evaporation fields of atoms surrounding voids compared with the matrix control the pattern of local atomic density variations in the APT reconstruction. In the following discussion, we will show how the different evaporation fields near the void can modify the shape of the specimen tip and their effects on local density variations.

## Discussion

It is commonly assumed that the specimen tip maintains a hemispherical shape during the entire APT experiment. The severely skewed void shapes and densities shown in the APT reconstructions (Fig. 3) indicate that the tip likely deforms from the ideal shape as evaporation proceeds through the void. Local changes in curvature cause aberrations in the magnification and ion flight trajectories, leading to the observed atomic density variations. To understand the complex effects introduced by the presence of voids, it is helpful to review how field-evaporation and reconstruction proceed for a perfect APT specimen. The

electric field ($E$) on the hemispherical surface of the specimen tip is calculated by

$$E = V/k_f r, \qquad (2)$$

where $V$ is the applied voltage, $r$ is the tip radius, and $k_f$ is a constant shape factor[11]. Surface atoms are ionized and emitted when $E$ is large enough to break the bonds between the atom and the surface. The evaporated atom follows a trajectory from its initial position on to the position-sensitive detector. To perform a tip reconstruction, the original coordinates of an emitted ion ($x$, $y$, $z$) are calculated using the following equations:

$$x = \frac{x_d}{\eta} = \frac{x_d}{L}\xi r, \qquad (3)$$

$$y = \frac{y_d}{\eta} = \frac{y_d}{L}\xi r, \qquad (4)$$

$$z = \Delta z + z', \qquad (5)$$

where $x_d$ and $y_d$ are the coordinates of the emitted ion hitting the detector; $\eta$ is the microscope magnification, which is equal to $L/\xi r$, where $L$ is the distance between the tip and the detector and $\xi$ is the compression factor. The order in which the ion hits the detector determines its $z$-coordinate. As written in Eq. (5), each ion's $z$-coordinate is calculated by adding a small increment, $\Delta z$, to the $z$-coordinate of the last evaporated atom ($z'$). Within one reconstruction, $L$ and $\xi$ are constants, $x_d$ and $y_d$ are recorded by the position-sensitive detector, and $r$ is estimated using Eq. (2) by assuming $E$ is equal to the evaporation field of the specimen material. Therefore, the tip radius will increase steadily as the applied voltage $V$ increases steadily[11].

The above assumptions and algorithms work very well for single-phase materials with no defects; however, they may not be valid for samples containing structural heterogeneities like voids[15,37], because the tip shape and radius will change abruptly near the heterogeneities. Based on insights obtained from simulations, we are able to illustrate the evolution of the tip shape as the evaporating surface passes through the void encapsulated within a segregation shell. Figure 5 displays cross-sectional snapshots during the simulated void evaporation with the low-field case on top and the high-field case on the bottom. A schematic model based on the simulation is shown in Fig. 6. The black lines delineate the tip cross-section; the circle with radius $R$ represents the void before evaporation; the colored spherical shell represents the chemical segregation layer (blue indicates a lower evaporation-field shell compared to the matrix and red indicates a higher evaporation-field shell).

In the low-field case, the specimen tip maintains a hemispherical shape with radius $r_t$ when the evaporating surface is far from the void (Fig. 6a). However, two obvious changes occur as the evaporating surface starts passing through the void (Fig. 6b). First, because the shell has a lower evaporation field, it will evaporate at a faster rate than the matrix until the shell surface curvature ($1/r_s$) decreases compared with the rest of the surface ($1/r_t$). Second, the opening of the void creates a hole having a diameter $D_v$ with a local radius $r_v$ that is much smaller than $r_t$ (Fig. 6b). According to previous studies[38], $r_v$ can be estimated using the following equation:

$$r_v \sim \frac{r_t}{2\left(\frac{r_t}{D_v} - 1\right)}. \qquad (6)$$

Note in the 2D schematics, it appears that there are only two sharp apexes with radius $r_v$, but because the tip has azimuthal symmetry in 3D, the real structure at the top of the void is a ring that can be described by an infinite number of sharp apexes. Both changes will significantly modify the normal evaporation and

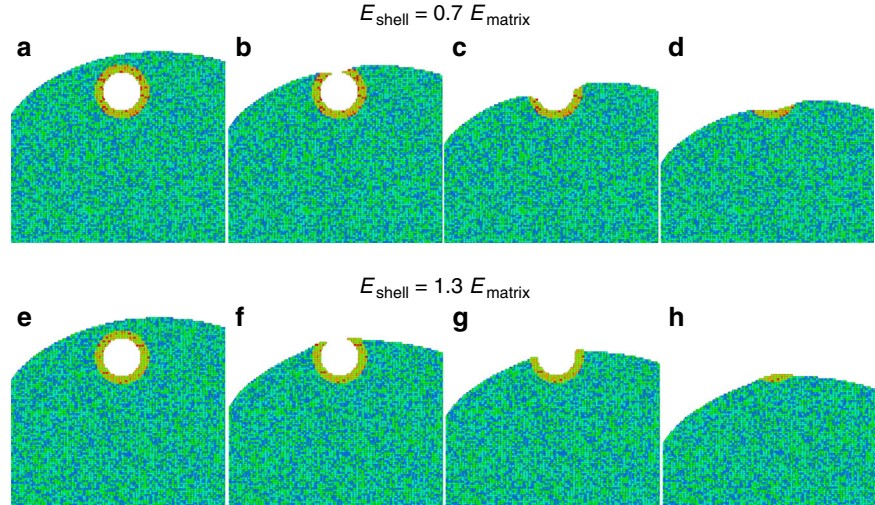

**Fig. 5 Cross-sectional snapshots of the simulated tip at different evaporation stages of the nanovoid. a–d** and **e–f** show the lower and higher field shell cases, respectively. The atoms in the matrix are colored in blue and dark green, whereas atoms in the shell are colored in red, yellow and light green.

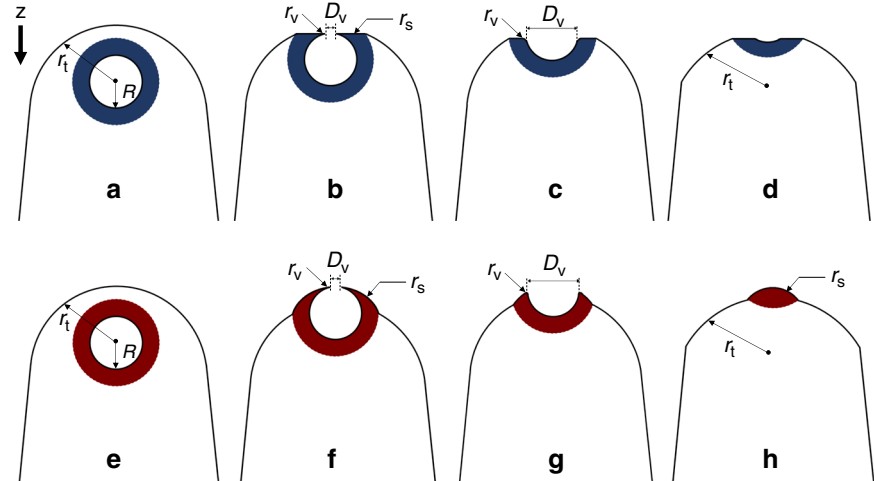

**Fig. 6 Schematic showing shape evolution of an APT tip near a void during the evaporation process. a–d** show a void with a lower evaporation-field shell colored in blue; **e–h** show a void with a higher evaporation-field shell colored in red.

reconstruction processes. First, as indicated by Eqs. (3) and (4), the much smaller $r_v$ will lead to a large lateral expansion of the ion positions near the top of the void, i.e., the ions will be projected into what should be empty void space on the detector. In addition, the overlap of emissions from the many ring apexes ($r_v$) will result in a higher atomic density in the center of the empty ring. In fact, such an overlapping mechanism is commonly observed in ring-shaped illumination devices with one example is presented in Supplementary Fig. 7. Second, the lower surface curvature near the segregation shell will compress the flight path of shell atoms and introduce an increase in local atomic density, which is similar to observations near a low-field precipitate presented in previous studies[39]. Therefore, the synergistic effects of the overlapping mechanism and the lower surface curvature generate the abrupt density increase in the λ shape shown in Figs. 2a and 3d. As evaporation continues, the ring apex ($r_v$) becomes blunter and the ring diameter $D_v$ increases; the overlapping mechanism gradually weakens (Fig. 6c) and the atomic density declines from its peak value. Near the end of evaporation through the void, the tip returns to a normal hemispherical shape

and the local atomic density decreases to the same level as in the alloy matrix (Fig. 6d).

Similar to the low-field case, for a void with a high-field shell, a ring with a local radius, $r_v$, is generated when the void opens and the overlapping mechanism fills the void empty space with atoms from the surrounding shell. However, unlike the low-field case, because the shell has a higher evaporation field than the matrix, the slower evaporation results in a higher surface curvature ($1/r_s$) and a higher local magnification near the shell (Fig. 6f). The higher local magnification will project atoms onto a larger region of the detector and result in a decrease in atomic density[39]. Therefore, despite the overlapping mechanism, the overall atomic density decreases near the top of the void. Because the ring apexes ($r_v$) are sitting on the high-curvature shell ($r_s$) that protrudes from the tip surface, the ring evaporates with the shell such that the high-curvature shell quickly diminishes (Fig. 6f–g). At this point, the overlapping mechanism starts to play a role and introduce an increase in the local atomic density as observed in the middle of the ω shape in Figs. 2b and 3f. Near the end of evaporation through the void, a high surface curvature ($1/r_s$) appears again

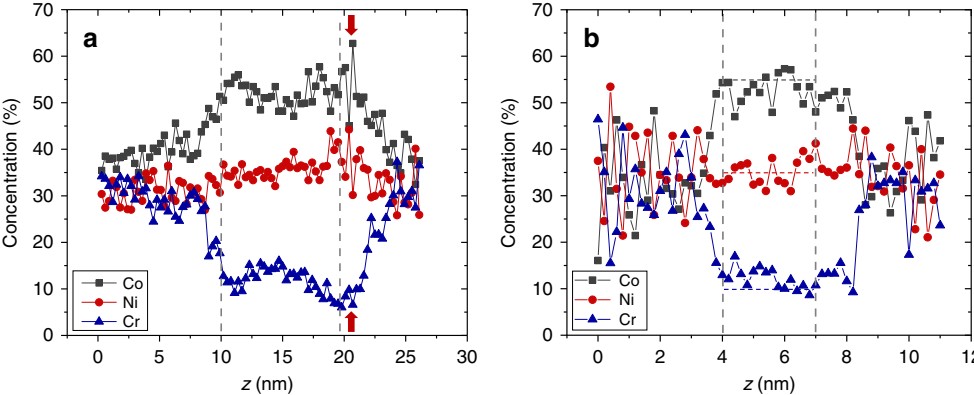

**Fig. 7 1D concentration profiles across voids in NiCoCr. a** is based on the experimental APT reconstruction and **b** is based on the simulated APT reconstruction. Horizontal dashed lines in **b** indicate the ground truth elemental concentrations in the simulation. Profiles were obtained by calculating element concentrations in 0.3 nm-wide bins from cylindrical ROIs passing through the void. As the simulated APT tip size is smaller than the experimental APT tip size, the x axis of experimental and simulated profiles cannot be compared directly. More details related to dimension of the simulated APT tip is provided in Methods section.

(Fig. 6h) due to the higher evaporation field of the shell and results in a decrease in local atomic density. This density variation disappears until the shell completely evaporates.

Based on the analyses described above, we demonstrate that the observed local density variations near voids in an APT dataset are controlled by the ion trajectory aberrations, which are introduced by the unique ring structure generated when voids open and evaporation-field differences between the segregation shell and the matrix. More details related to the ion trajectory aberrations and ion crossing revealed by our simulations are provided in Supplementary Fig. 8. An animation showing the evolution of the tip surface during the void evaporation are provided in Supplementary Movie 3.

The point of this study is to understand how the aforementioned ion trajectory aberrations and variations in local density affect the measured chemical composition near voids in APT data. Simulations provide a powerful approach to answer this question as we can set the simulated shell compositions to the experimental values and track the flight trajectory of each atom. A comparison of the 1D concentration profiles acquired across a void from both the APT experiment and the simulation for NiCoCr is shown in Fig. 7. The vertical dashed lines indicate the void location along the profile. As shown in the experimental line profile (Fig. 7a), Co is enriched and Cr is depleted near the void, with an average Ni, Cr, and Co of $36 \pm 3$ at.%, $12 \pm 2$ at.%, and $52 \pm 3$ at.%, respectively. Very similar concentration profiles can be obtained using the simulated reconstruction (Fig. 7b), assuming a segregation shell surrounding the void with assigned concentration values listed in Table 1. The qualitative agreement between the experimental and simulated concentration profiles, as well as the agreement between the simulated and assigned concentration values, indicate that APT can provide a relatively accurate measurement of the chemical composition around voids despite the aberrated ion trajectories. Using the simulation, we are able to extract the origin of atoms along the concentration profile in both the low- and high-field cases. As shown in Fig. 8a, near voids with low-field shells like in NiCoCr, most atoms are from the segregation shell and only 10% or fewer are mixed with atoms from the matrix, suggesting that intermixing induced by the ion trajectory aberration is relatively low and explains why a relatively accurate segregation measurement can be achieved using APT. Zero intermixing is observed in the high-evaporation-field case within the void (Fig. 8b), which shows that a compositional measurement of the shell can be obtained with a higher accuracy in this case. One interesting observation in Fig. 8a is that the

**Table 1 Chemical composition in Co, Ni, and Cr (at.%) of the simulated matrix and shell.**

| Structure | Co | Ni | Cr |
|---|---|---|---|
| Matrix | 33.3 | 33.3 | 33.4 |
| Shell | 55 | 35 | 10 |

bottom region of the void exhibits the lowest extent of intermixing from the matrix (~4%), indicating that the most accurate segregation measurement should be made in this region. This is reasonable because near the end of the evaporation through a void, the ring apex ($r_v$) diminishes and the tip recovers to a more ideal hemispherical shape, thereby minimizing the ion trajectory aberrations. Correspondingly, the experimental Cr concentration reaches a minimum (~$8 \pm 2$ at.%) and the Co concentration reaches a maximum (~$54 \pm 5$ at.%) near the bottom of the void, which is between 19–21 nm (marked by red arrows in Fig. 7a). These values should be closest to the actual radiation-induced segregation near a void. Compared to the EDS segregation measurements, which shows 26 at.% Cr and 38 at.% Co near the void in the NiCoCr sample (Supplementary Fig. 1), we can conclude a much more accurate segregation measurement is achieved using APT. Note that the simulated density variations and concentration profiles do not match exactly with the experimental results, which is likely due to the different void sizes, tip radii, and segregation shell widths between the experiment and simulation. We also found the magnitude of density variations relies on the void size relative to the specimen tip radius. The effects of void size on composition measurement will be explored in future research.

Based on this study, we propose following a general practice for interpreting APT data containing voids. Local atomic density variations identified by iso-density surfaces can be used to locate the void positions in the APT reconstruction. To confirm that the density variations are introduced by voids rather than other structural heterogeneities such as precipitates, the shape of the density variations shown in 1D density profiles or 2D density contours can be checked with the evaporation field indicated by local ion charge state ratios. If the density variation is from a void, a λ-shaped density variation should originate from a region with a lower charge state ratio and a lower evaporation field than the matrix, and an ω-shaped variation should originate from a region with a higher charge state ratio and a higher evaporation field.

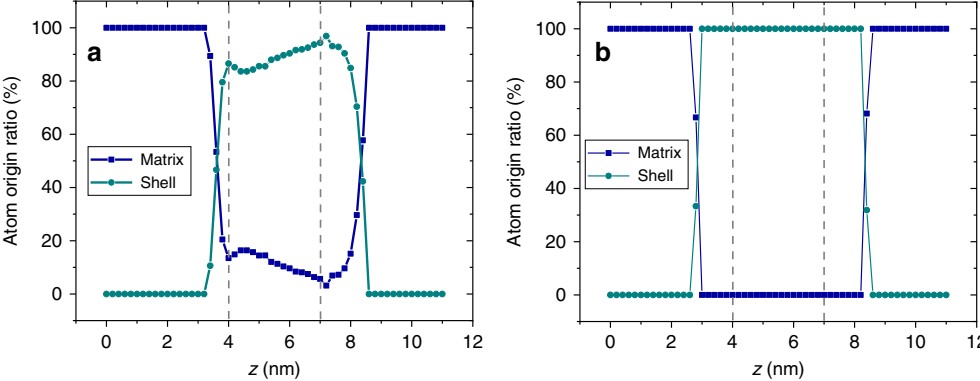

**Fig. 8 Origin of atoms along concentration profiles based on simulated APT reconstruction. a** is for a void with a low-evaporation-field and **b** is for a void with a high-evaporation-field segregation shell.

With this confirmation, an accurate composition measurement near voids can be obtained using the APT data directly. This can be done by creating a 1D concentration profile perpendicular to the reconstructed hemisphere that travels through the void and measuring the compositions toward the bottom of the void and into the matrix. In addition, the dimensions of the density variation region can provide a reasonable estimation of the void size, as supported by our correlative experiment (Fig. 2c–d). It is worth mentioning that the current work aims at illustrating the physical mechanisms for void evaporation in APT and demonstrating a workflow to acquire an accurate concentration measurement near nanovoids. To fully quantify the relation between nanovoids and the ion trajectory aberrations, future studies are still necessary to explore a much larger parameter space like void size, shape, density, etc. The readers are always encouraged to take a similar approach that combines both the APT experiment and simulations to get a more quantitative interpretation of their APT data containing voids.

In summary, correlative APT-STEM studies show that contrary to the notion of being an empty structure, a nanosized void can introduce either an increase or decrease in local atomic density in an APT reconstruction. We have discovered that the shape of the density variations created during field-evaporation through a void are controlled by two mechanisms: overlapping due to the unique ring structure associated with the void and the different surface curvatures generated by the different evaporation fields of the void segregation shells compared to the matrix. Based on the insights obtained in this study, the chemical compositions near voids can be measured using APT with a much higher accuracy than STEM-based EDS/EELS methods, which is important for material characterizations in a number of research fields.

## Methods

**SP-CSA fabrication, ion irradiation, and conventional TEM characterization**. The SP-CSAs were synthesized by arc-melting and drop-casting[40]. The SP-CSAs were confirmed as single-crystal solid-solutions in the face-centered cubic (fcc) structure using Laue X-ray backscatter diffraction[26,40]. The component elements (Ni, Co, Cr, Fe) were carefully weighed based on the designed composition for the arc-melting process, and the doped elements (Al and Si) were controlled to be less than 1 wt%.

Helium bubbles were generated using He ion implantation and voids were generated using Ni ion irradiation. The 200 kV Danfysik Research Ion Implanter at the Ion Beam Materials Laboratory at Los Alamos National Laboratory was used for He implantation in Ni, NiFe, and NiCoCr. In all, 200 keV He ions were implanted into the bulk samples at 500 °C at a beam flux of ~2 × 10$^{13}$ ions/cm$^2$s. The total irradiation fluence was 5 × 10$^{16}$ ions/cm$^2$. Nickel implantation in doped NiCoCrFe was conducted in the Ion Beam Materials Lab at the University of Tennessee-Knoxville with 3 MeV Ni ions at a beam flux of 2.8 × 10$^{12}$ ions/cm$^2$s to a fluence of 8 × 10$^{16}$ ions/cm$^2$ at 500 °C[41].

In order to get a preliminary understanding of the void distribution and chemical segregation near the voids, thin foils for transmission electron microscopy

(TEM) and STEM analysis were prepared using normal lift-out procedures in an FEI Nova 200 focused-ion beam (FIB) instrument[42]. Conventional TEM analyses were performed to characterize the void sizes and densities in the SP-CSAs using an FEI Titan TEM/STEM operated at 300 kV. As shown in Supplementary Fig. 9, the void diameters ranged from 4 to12 nm and the void density ranged from 0.6 × 10$^{-5}$ nm$^{-3}$ to 2.9 × 10$^{-5}$ nm$^{-3}$. The relatively high density and small sizes of the voids make it feasible to capture a few voids within the volume of the needle-shaped APT specimen, which is typically a few hundred nanometers long and tens of nanometers in diameter. STEM-based EDS characterization of chemical segregation near voids was conducted in NiCoCr samples using the high-solid-angle EDS system within a 200 kV FEI Talos F200X microscope.

**Correlative APT-STEM analysis**. To conduct the correlative APT and STEM study, a bronze specimen post, which can be installed both in the puck holder for APT analysis and in the Fischione on-axis rotational holder (Model 2050) for STEM analysis was prepared. The process for preparing the specimen post is discussed in detail in a previous study[43]. Following the normal FIB procedures for APT specimen tip preparation[44], a wedge-shaped lamella was lifted-out from the irradiated sample surface and mounted onto the bronze specimen post. A series of annular FIB millings generated a <120 nm diameter needle-shaped specimen tip, making the tip transparent to the 300 keV electron beam. The tip was then installed in the Fischione on-axis rotational holder and transferred into the Titan microscope for STEM analysis. This rotational holder allows for a full 360° on-axis rotation of the specimen. For our analysis, the specimen was rotated over a 180° range and one HAADF-STEM image was acquired for each 5° rotational increment, totaling 37 collected HAADF-STEM images from the specimen. Combining the HAADF-STEM images at different rotation angles, the size and coordinates of each void were determined. During the STEM analysis, carbon contamination from the microscope column concentrates on the specimen surface, which can affect the following APT experiment[43]; therefore, gentle cleaning using a 2 kV Ga beam in the FIB was applied to the specimen to remove the possible contamination layer after the STEM analysis and before the APT experiment.

APT relies on the specimen to be shaped into a conical frustrum with a hemispherical cap that has a radius on the order of ~50 nm. An electric potential between the specimen and a local electrode create a high enough electric field at the hemispherical sample surface to ionize and field evaporate surface atoms, i.e., field evaporation. The ions are projected onto a position-sensitive detector such that the hemispherical surface is magnified to ~×1,000,000. Voltage or laser pulses allow for the mass-to-charge ratio of the ionized atoms to be determined using time-of-flight (ToF). The X and Y detector positions, ToF, and the order of ion evaporation recorded from the experiment can be used to reconstruct the real space x, y, and z positions of the detected ions using a computer algorithm. A simplistic way to think about APT is that the specimen surface is projected and imaged atom-by-atom in a destructive manner such that each surface is removed for the next surface to be imaged. In our study, a CAMECA LEAP 4000X HR system was used for APT analyses. The specimen was run in laser mode at 30 K with a pulse repetition rate of 200 kHz, a detection rate of 0.004 atoms per pulse, and a 70 pJ laser energy. About 14–25 million ions were acquired for each APT tip. In order to demonstrate that the results were independent of the APT-operating conditions, APT tips were prepared using the NiCoCr sample and were run using the voltage mode as well at 56 K with a 200 kHz pulse repetition rate, a detection rate of 0.002 ions per pulse, and a 30% pulse fraction. All the collected APT data were reconstructed and analyzed using CAMECA's integrated visualization and analysis software (IVAS) 3.8.0. For the reconstruction, the detector efficiency was set as 0.36, the image compression factor was set as 1.65 and the sphere-cone-radius ratio was set to have tangential continuity. For each APT tip, an image was acquired using a scanning electron microscope (SEM) before the APT experiment. The SEM image was then used for the reconstruction in the tip profile mode provided by IVAS. For the IVAS

analysis, the voxel size and delocalization were set as 1 nm and 3 nm in all directions, respectively. It is worth noting that in the APT analyses, the local atomic density of an ROI is calculated as the number of reconstructed atoms in the ROI divided by the ROI volume. Therefore, this atomic density only presents the spatial distribution of the reconstructed atoms, which may not necessarily represent the real positions of the original atoms in the materials. The readers should not regard the atomic density discussed in this work as an intrinsic property of the materials.

For APT analysis of He bubbles, despite the presence of He inside the bubble, no He peak could be identified in the APT mass spectrum. This can be explained by the pressurized gas state of He inside the bubbles, the pressure of which were found as high as a few GPa in previous studies[30,45].

For the correlative STEM-APT results shown in Fig. 1, it is worth noting that there are a few small high-density regions in the APT reconstruction that do not have corresponding voids in the HAADF-STEM images, probably because the voids are so small that they are difficult to observe in the HAADF-STEM images. To show that the observed characteristic density variation near voids is not a specific case for certain voids, 1D density profiles across almost all the voids shown in Fig. 1e are provided in Supplementary Fig. 10. A common λ shape can be found in all the density profiles.

**Procedures for simulated APT experiment**. The tip material for the simulated APT experiment was in the cubic structure, which is different from the fcc structure of the studied SP-CSAs and Ni, but a previous study suggests crystallography only plays a second-order role in measured APT artifacts[46]. The tip was defined with a small radius ($r$) ~ 15 nm at the apex of a truncated cylinder. The void was defined by a geometrically spherical region with no atoms inside. To generate comparable simulated atom distributions with respect to the experiment, the aspect ratio of the void size with respect to the tip radius in the simulation was set to be close to the experimental values[16]. In the simulation, the void radius was set to $0.1r$, which is consistent with the experimental observation (as shown in Figs. 1a and 2a, the experimentally observed tip radius $r$ is ~ 40–50 nm and void radius is ~ 4.6 nm). In order to compare the simulated and experimental composition profiles, the matrix of the tip was described with three types of atoms (Ni, Co, and Cr) randomly distributed with the same composition as the experimental specimen. The element concentration in the segregated shell was also set to values similar to the experimental measurement results as shown in Fig. 7a. The elemental concentrations of the simulated tip matrix and the segregated shell are summarized in Table 1. The thickness of the segregated shell ($e$) was set such that the $R/e$ ratio was equal to 2.25, which is also close to the experimental value (4.6 nm/2 nm). It is worth mentioning that because of the limited available computing resource, the size of the simulated tip is smaller than the size of the experimental tip. Therefore, the dimension of the simulated void cannot be directly compared with the dimension of the void observed in the experiment. However, all the variations in local atomic densities (Figs. 2 and 3) as well as the element concentration profiles (Fig. 7) near the void were successfully reproduced by our simulations.

## Data availability

All data generated or analyzed during this study are included in this published Article and its Supplementary Information files. Further information is also available from the corresponding authors upon reasonable request.

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

## Acknowledgements

This work was supported by the Energy Dissipation to Defect Evolution (EDDE) Center, an Energy Frontier Research Center funded by the US Department of Energy (DOE), Office of Science, Basic Energy Sciences under contract number DE-AC05-00OR22725. Electron microscopy and APT were conducted at Oak Ridge National Laboratory's Center for Nanophase Materials Sciences (CNMS), which is a US DOE Office of Science User Facility. Helium implantations were supported by the Center for Integrated Nanotechnologies (CINT), a DOE Office of Science user facility jointly operated by Los Alamos and Sandia National Laboratories. Nickel irradiations were performed at the Ion Beam Materials Laboratory (IBML, https://ibml.utk.edu/) located on the campus of the University of Tennessee, Knoxville. We acknowledge the financial support of the Region Normandie–FEDER and ANR / EMC3 Labex, DYNAMITE project and support from Semiconductor Research Corporation (SRC) under task ID 2679.001 for the field-evaporation simulation. We thank James Burns for assistance with sample preparation and running the APT experiments.

## Author contributions

X.W. and J.D.P. wrote the manuscript. J.D.P., X.W., K.L.M, W.J.W., and Y. Z. conceived the project. X.W. and J.D.P. designed the experiments. C.H. and F.V. designed and conducted the field-evaporation simulation. X.W., J.D.P., C.H., F.V., and B.G. provided theoretical explanations to the experiment and simulation results. X.W. and J.D.P. performed the correlative APT and STEM analysis. X.W. and B.S. conducted the EDS and EELS experiments. X.W. and Z.F. conducted the TEM characterization of voids. W.G. conducted part of the APT analysis. K.J. and H.B. prepared the materials. D.C., Y.W., and Z.F. conducted the irradiation experiments. All authors discussed and commented on the manuscript.

## Competing interests

The authors declare no competing interests.
