## [Peer Review File · Nature Communications]

Reviewers' comments:

Reviewer #1 (Remarks to the Author):

A review for "Interpreting nano-voids in APT data using a correlative APT, STEM, and simulated APT experimental approach" by X. Wang et al.

Summary:

This manuscript investigates the mechanism by which nanovoids are imaged in a 3D atom probe microscope. In order to have a clear picture of how the atoms are removed and projected on a detector the manuscript brings together atom probe data, STEM data, and simulations of atom probe data. The authors find that for their samples, the atom probe data can be well described by simulations taking into account the shape/size of the voids, the chemical segregation around the voids, and an effective evaporation field around the voids. Specifically, they demonstrate a low-field shell in high-field matrix and a high-field shell in a low-field matrix give characteristic density variations (and charge state ratios) which may be used to identify the presence of a void and then to help extract out information about the void and any elemental segregation near the void.

Review:

I enjoyed the opportunity to read the manuscript and I found it to be well written overall. It brings together several techniques (experimental and theoretical) to gain insight into an artifact in atom probe data sets that contain nanovoids. I have a number of technical comments/critiques outlined below, but I would think that the authors can address them with appropriate revisions. However, I do have some reservations about the broader impact of the work and how well it fits into Nature Communications. The authors acknowledge that the atom probe community already knows that voids are not imaged in atom probe data accurately because of trajectory aberrations due to the non-spherical topography of the tip as it runs through the voids (refs 17-21 from the manuscript). From my reading, the author's present contribution is to give some insight into the exact nature of those trajectory aberrations for a particular material (SP-CSAs). The authors give some general guidance on how their results may be applied to future data sets, but it is unknown how accurate this extrapolation will be since they have not been able to explore a significant fraction of the parameter space (the computational work is quite intensive I would assume). To me this seems to be an incremental (albeit well-done and very worthwhile) contribution to a specific field and it is difficult to see how that would benefit from the readership of Nature Communications. In my view, this work is more appropriate for a microscopy focused journal but not for Nature Communications. Having said that, I will defer to the authors' and editors' view on whether Nature Communications is an appropriate journal. Based on its technical merits I would recommend the manuscript be accepted pending revisions.

Manuscript comments:

- In the abstract the authors state that the "composition can be determined [by APT] with a higher accuracy than STEM-based techniques. However, it is not clear that their data support such a conclusion as strongly as it is stated. First off, no significant attempt was made to quantify the composition accurately with STEM-EDS. If they measured the thickness of the STEM-EDS sample, and created a model of the void similar to the one they used in the atom probe simulation and then iteratively simulated the EDS measurement they could plausibly get reasonable agreement. It seems to me that equal effort should be put into the STEM-EDS measurement as the APT measurement if that claim is to be made. Furthermore, given the assumptions used in the void model (discrete shell with a certain thickness and a constant composition) it is unclear if the simulations are fine-grained enough to ensure the atom probe data is 'quantitatively' accurate. Please either remove the claim about STEM-EDS, or support it with more careful analysis.
- The voids in Figure 1e-h are quite small on the page and difficult to see – there is currently a lot of white space on the figure that could be used to show the reconstructions enlarged by 2X for viewing. Please enlarge these images to a size where the reader can more easily interpret them.
- While several voids in Figure 1 do resemble a downward pointing triangle, many do not – is this a

camera viewing angle issue? One thing that is difficult to judge from this manuscript is how much variation/noise there is on the lambda and omega characteristic shapes. For example, if you were to plot the density profile for every void in the dataset from Figure 1, would they all be unambiguously lambda shaped? Showing some minimally biased statistics that can assure the reader that the result is robust would be helpful.

- How sensitive are the characteristic density profile shapes to reconstruction parameters/models?
- Why does the void not appear to have a spherical bottom in Figure 3d?

- The simulations were performed at a smaller length scale than the experimental voids. Do you anticipate any particularly scaling relationships to hold such that you can extrapolate from one size scale to another? To play devil's advocate, one might argue that the volumes scale as r^3 and the shell surface area scales as r^2 so wouldn't that potentially affect the evaporation behavior (making it harder to extrapolate the simulations)?

- Where did the 30% higher/lower evaporation field numbers come from? How sensitive are the results to these numbers? Some justification should be given for the choice.

- Line 242 you state that the "cases prove that the simulations quantitatively reproduced the experiments". I strongly disagree. The results are not quantitative. If they were quantitative then you should plot them on the same axes. But the y-scale of the data and simulation are quite different, and the x-scale of the data and simulation are quite different. The results match up qualitatively quite well, but in no way do they appear to match up quantitatively. Please remove such language.

- Figure 6 is nice in that it gives a simple interpretation for the results of the simulation. However, the reader has not had a chance to see the results of the simulation directly. Is it possible to show some cross sectional (x,z view) snapshots from the simulation at various points? Basically just the (x,z) view of the data shown in Figure S6. You could leave it in the supplementary, but it would be nice to see the data that the schematic in Figure 6 is derived from.

- Line 376 you again state that the simulation "quantitatively reproduced" the data. I again strongly disagree that the agreement is quantitative. It appears qualitative to me. Please remove such language. Perhaps you could plot the "ground truth" compositions that you built into the simulation on Figure 7b to highlight that the shell compositions are not strongly altered despite the trajectory aberrations. That may make your point in a more convincing manner.

- The data broadly convince me that for a selected evaporation field, shell/matrix composition, and shell/void size, that the concentration observed at the bottom of the void is close to the "real" value, but do you have any idea how much this agreement may change by changing these parameters? Right the claims you make about voids are implied to be relatively broad when only a very small region of parameter space has been explored in simulations and these parameters are not entirely the same as for the experiment. I'd like to see much more emphasis on the qualitative nature of the comparison for your material system – which is quite good and compelling and adds to the knowledge based of the atom probe field – and less on the 'quantitative' nature (which is lacking in my view).

Supporting comments:

- Figure S1 is difficult to interpret. The voids in the HAADF images from Figure 1 appear roughly spherical, but the chromium deficient regions of the STEM-EDS image do not appear spherical. A discussion of this discrepancy would be useful here. Perhaps including the HAADF image for the same region as Figure S1a would help.

- Why do the profiles in Figure S4 look so different from those in Figure 7a? Especially the cobalt – the shape looks qualitatively different.

- Figure S5. I think I understand the point you are trying to make here, but not enough detail is given for a real understanding of what the inset is trying to show. Is the appearance of the 'ring-shaped illumination model intensity plot dependent on the ratio of the gaussian width parameter relative to the circle diameter?

- The word 'radical' was used in many places instead of 'radial'

- A circular colormap might make Figure S6a/c a easier to interpret.

Reviewer #2 (Remarks to the Author):

This paper reports on the interpretation of nanovoids in APT reconstruction using a correlative approach, which is an important issue for several applications.

Overall, I have nothing to criticize about the work, which makes it difficult for me to write a constructive review that could help to further improve the manuscript.

I only have a few minor points:

- When you used voltage mode to perform the experiments, you didn't detect any He⁺ species? Even though the He pressure corresponds to a few GPa, some He could be ionized. But I guess the background noise of the mass spectra doesn't allow to observe it.
- You claimed that you didn't find any experimental counterpart for the equal-field case (line 240). It is a bit surprising since you analyzed pure Ni (with voids), and I assume that the shell surrounding the void is also made of Ni, but the sigma explanation is reasonable.
- It would be good to discuss the domain of applicability of your work. The voids are 4 to 12 nm, but is your approach applicable to bigger voids? Especially, I was thinking of nanoporous Au, with pores in the 20-50 nm size range, that have been filled with Co and Fe filling by e-beam induced deposition (Microsc. Microanal. 2015, 21, 557-563), or filled with Cu by electrodeposition (Mater. Charact. 2017, 128, 269-277 / Electrochim. Acta 2018, 283, 611-618), and also nanoporous Si filled with Ni (Ultramicroscopy 2017, 182, 112-117). Is your approach applicable to such systems?
- Could you also briefly discuss the applicability in the case of non-perfectly round-shaped voids?
- On line 304, you say that r is estimated using Eq(1). I think it should be Eq(2) instead.
- On lines 446 and 449, there was an issue in the conversion of the pdf, and the unit of 500 appears as a square.
- In the SI, line 45, you wrote LEAP 4009X HR, I think it should be 4000.

Reviewer #3 (Remarks to the Author):

The presented study is an extremely well executed piece of work that goes to great lengths to get to understand the chemistry of nanovoids in materials as well as possible with currently available methods. I highly recommend the publication of this article. Especially, this is one of the first uses of full 3d field evaporation simulation to aide the understanding of a real word problem in APT.

I nevertheless have a few comments about necessary improvements to the manuscript. Most importantly, since so much effort has gone into this work, it is incredibly dense to read. This is no problem for the few fully initiated into the topic, but for the general APT community it would be very helpful to illustrate the expected image artefacts and how they would impact a trajectory (Bas type) reconstruction in the introduction of the paper. This happens somewhere half way through, where you would already have lost many readers.

In the simulations, a deviation in the evaporation field of 30% with respect to the matrix is assumed. How is this justified? for many d-block metals, the (pure elemental) evaporation fields differ by much less than that. Since much of the shape of the resulting concentration profiles is dependent on when a "crossover point" of the ion trajectories is reached, surely there would be a development of the observed concentration profiles dependent on the differences in evaporation field.

Lastly, I am a bit disappointed that figure 6 only contains schematic drawings, since from the simulations, the field evaporation forms at any step during the field evaporation of the nanovoid are available. Pls consider putting cross sections of the nano-voids into the figure, maybe together

with the schematics in a left-right half/half, or entirely.

Nevertheless, this is great work and absolutely deserves publishing.

Response letter

Journal of Submission: Nature Communications

Title: *Interpreting Nano-voids in Atom Probe Tomography Data for Accurate Local Compositional Measurements*

Authors: *Xing Wang, Constantinos Hatzoglou, Brian Sneed, Zhe Fan, Wei Guo, Ke Jin, Di Chen, Hongbin Bei, Yongqiang Wang, William J. Weber, Yanwen Zhang, Baptiste Gault, Karren L. More, Francois Vurpillot, and Jonathan D. Poplawsky*

Letter to the reviewers:

We are grateful for the reviewers' time in preparing a detailed and insightful review. It is encouraging to us that all reviewers speak highly of the technical merits of this work. Since this study provides an efficient approach for accurate characterization of nanovoid chemistry, we believe it will be interesting to many readers of Nature Communication. We have responded to the reviewers' comments and revised the manuscript accordingly. For clarity, the reviewer's comments are italicized and our responses to each comment follow a bold "Response" title. A Microsoft Word document with modified parts marked in red is attached. Also, several additional supplementary figures were added to the supplementary information to clarify points brought up by the reviewers.

We are confident that we have addressed all of the reviewers' comments and we hope that the manuscript is now acceptable for publication.

Reviewers' comments:

Reviewer #1 (Remarks to the Author):

A review for "Interpreting nano-voids in APT data using a correlative APT, STEM, and simulated APT experimental approach" by X. Wang et al.

Summary:

This manuscript investigates the mechanism by which nanovoids are imaged in a 3D atom probe microscope. In order to have a clear picture of how the atoms are removed and projected on a detector the manuscript brings together atom probe data, STEM data, and simulations of atom probe data. The authors find that for their samples, the atom probe data can be well described by simulations taking into account the shape/size of the voids, the chemical segregation around the voids, and an effective evaporation field around the voids. Specifically, they demonstrate a low-field shell in high-field matrix and a high-field shell in a low-field matrix give characteristic density variations (and charge state ratios) which may be used to identify the presence of a void and then to help extract out information about the void and any elemental segregation near the void.

Review:

I enjoyed the opportunity to read the manuscript and I found it to be well written overall. It brings together several techniques (experimental and theoretical) to gain insight into an artifact in atom probe data sets that contain nanovoids. I have a number of technical comments/critiques

outlined below, but I would think that the authors can address them with appropriate revisions. However, I do have some reservations about the broader impact of the work and how well it fits into Nature Communications. The authors acknowledge that the atom probe community already knows that voids are not imaged in atom probe data accurately because of trajectory aberrations due to the non-spherical topography of the tip as it runs through the voids (refs 17-21 from the manuscript). From my reading, the author's present contribution is to give some insight into the exact nature of those trajectory aberrations for a particular material (SP-CSAs). The authors give some general guidance on how their results may be applied to future data sets, but it is unknown how accurate this extrapolation will be since they have not been able to explore a significant fraction of the parameter space (the computational work is quite intensive I would assume). To me this seems to be an incremental (albeit well-done and very worthwhile) contribution to a specific field and it is difficult to see how that would benefit from the readership of Nature Communications. In my view, this work is more appropriate for a microscopy focused journal but not for Nature Communications. Having said that, I will defer to the authors' and editors' view on whether Nature Communications is an appropriate journal. Based on its technical merits I would recommend the manuscript be accepted pending revisions.

Response: We appreciate the reviewer's helpful comments and would like to provide a few additional points to highlight the broad impact of this work.

1. Nanosized voids are common defect structures in alloys, semiconductors, and catalysts. In this work, Ni and Ni-based single-phase concentrated solid solution alloys (SP-CSAs) were chosen as a model system to investigate the field evaporation process near nanovoids, but our conclusions are independent from the studied materials. For any materials analyzed by APT, the field evaporation process is the same, i.e., the surface atoms will be ionized, extracted, and then projected to the detector. Therefore, the same aberrations will occur near nanovoids in different materials, and our proposed approach can be applied to nanovoids in various structural and functional materials. As one demonstration, we can change the evaporation field values of the simulated materials to match other crystalline materials, and the similar trend of density variations near nanovoids will still be observed.

SP-CSAs are a perfect model system for this study because there are obvious composition variations in the void shells, which label the shell ions from the matrix ions. In addition, SP-CSAs (including medium and high entropy alloys) are an important class of alloys with fast-growing interests from the material research community¹⁻³.

2. We would like to stress that this is the first comprehensive study revealing the nature of the field evaporation process near nanovoids by combining multiple techniques (e.g., correlative STEM-APT and field evaporation simulations). Although the existence of aberrations near nanovoids has been known within the APT community for a long time, no systematic study has been performed to explore the physical mechanisms and understand the effects of voids on the APT analysis before this work. Based on cross-validations between experiments and simulations, we unambiguously demonstrated that the cavity structure and the ratio of the shell and matrix evaporation fields determine the local density variations near nanovoids in APT. We also demonstrated how the density variations can be used to estimate void sizes and show the best method for void shell chemical quantification and the limitations with quantification. The

insights gained from this work set the stage for future APT studies on void-containing materials for years to come.

3. We also hope to mention that a substantial amount of parameter space has been examined in this manuscript. Specifically, our conclusions were testified and validated in four different materials, including Ni, NiFe, NiCoCr and doped NiCoCrFe. In the revised manuscript, we also added the simulations with a large variation in evaporation fields of the segregation shells (from 0.2 to 2 times electric field of the matrix) and confirmed our conclusions. Our ongoing research finds that the void shape and relative void size can affect the magnitude of local density variations, but the conclusions shown in this study are still valid. Nevertheless, we completely agree with the reviewer that future investigations are necessary to fully quantify the relation between the void structures and the ion trajectory aberrations. In this work, we focus on elaborating the physical mechanism of the atomic-density variations near nanovoids and promoting the APT method for accurate void chemistry characterization, which will lay the foundation for more technically detailed research in the future.

To highlight the broad impact of this work, we modified the title of the manuscript to “Interpreting Nano-voids in Atom Probe Tomography Data for Accurate Local Compositional Measurements”. We also added a brief discussion to the second to last paragraph to stress the impact and limitations of this study.

Manuscript comments:

1- In the abstract the authors state that the “composition can be determined [by APT] with a higher accuracy than STEM-based techniques. However, it is not clear that their data support such a conclusion as strongly as it is stated. First off, no significant attempt was made to quantify the composition accurately with STEM-EDS. If they measured the thickness of the STEM-EDS sample, and created a model of the void similar to the one they used in the atom probe simulation and then iteratively simulated the EDS measurement they could plausibly get reasonable agreement. It seems to me that equal effort should be put into the STEM-EDS measurement as the APT measurement if that claim is to be made. Furthermore, given the assumptions used in the void model (discrete shell with a certain thickness and a constant composition) it is unclear if the simulations are fine-grained enough to ensure the atom probe data is ‘quantitatively’ accurate. Please either remove the claim about STEM-EDS, or support it with more careful analysis.

Response: Based on the following three reasons, we think APT method provides a more accurate direct composition analysis near voids than STEM-based techniques like EDS or EELS.

1. Due to the limitation of detector efficiency, sample absorbing effect and other factors, it is very challenging for EDS/EELS to quantify doped elements with low concentrations. Based on our experience, usually EDS/EELS techniques can measure elements with concentrations as low as 1 at. %. Under certain optimized conditions, this chemical sensitivity may reach around 0.1 at. % (1000 ppm)⁴. In contrast, mass spectroscopy in APT provides a much higher chemical sensitivity. It is routine for APT to quantify elements with concentrations as low as 10 ppm⁵. Therefore, for characterizing low-concentration elements near voids (e.g., the doped Al and Si in NiCoCrFe

shown in this work), the proposed APT method is much more accurate than STEM-based EDS/EELS.

2. For elements with high matrix concentrations (e.g., Cr in $\text{Ni}_{0.33}\text{Co}_{0.33}\text{Co}_{0.33}$ in this our work), it is also difficult for EDS/EELS to reach a similar accuracy as APT, even with the model calculation as suggested by the reviewer. There are two major limitations. First, it is usually hard to obtain accurate absolute thickness values of the STEM thin foils. For common techniques like convergent electron beam diffraction and EELS spectra, the error in absolute thickness measurement is around 10%⁶⁻⁸. Second, we need to estimate the size of the segregation shell for the model calculation. This size can be hard to accurately estimate because of potential overlaps of multiple segregation shells along the electron beam direction and low sensitivities of EDS/EELS for small local concentration changes. Both limitations originate from the fact that when using EDS/EELS, we have to project the 3D samples onto a 2D plane for the measurement, so the information along the electron beam direction is missing. In contrast, APT is a 3D technique by its nature.

3. We completely agree with the reviewer that if more efforts are made, the accuracy of the STEM-EDS/EELS measurement can be future increased. Such efforts can be a model calculation as the reviewer suggested, or EDS/EELS tomography⁹. In certain optimized cases, these approaches may help EDS/EELS reach a similar composition accuracy to the APT method, but they will require a substantial amount of time, effort and additional calculations. Meanwhile, with the insights obtained from this study, an APT analysis can be completed in a few hours and the composition profiles near voids will be obtained. In this sense, we can say that the APT is a more accurate “direct measurement” of the void chemistry.

To be more rigorous, we modify our statement in the abstract as “composition can be *directly* determined with a higher accuracy than STEM-based techniques”.

For the reviewer’s comment on the void model for APT simulation: we completely agree that the current model is a coarse-grained approximation to the real condition. The actual concentration profiles near nanovoids are probably diffusive with gradients. With current coarse-grained approximation, our simulation has generated concentration profiles (Fig. 7b) in a good agreement with the experimental profiles (Fig. 7a). The agreement should be further improved if a finer-grain model is employed in our simulation. Here we use the agreement between the simulation and experiment to demonstrate that the APT method can provide a more accurate concentration measurement near voids than STEM-based EDS/EELS. We agree with the reviewer that we should put “more emphasis on the qualitative nature of the comparison”. In the revised manuscript, we removed all the “quantitatively” statement based on the reviewer’s comment 8 and 10.

2- *The voids in Figure 1e-h are quite small on the page and difficult to see – there is currently a lot of white space on the figure that could be used to show the reconstructions enlarged by 2X for viewing. Please enlarge these images to a size where the reader can more easily interpret them.*

Response: Fig. 1e-h have been enlarged based on the suggestion.

3- *While several voids in Figure 1 do resemble a downward pointing triangle, many do not – is this a camera viewing angle issue? One thing that is difficult to judge from this manuscript is how*

much variation/noise there is on the lambda and omega characteristic shapes. For example, if you were to plot the density profile for every void in the dataset from Figure 1, would they all be unambiguously lambda shaped? Showing some minimally biased statistics that can assure the reader that the result is robust would be helpful.

Response: Here we plot the density profiles of every void shown in Fig. 1 in the manuscript. Fig. r1a shows the void positions in the APT reconstruction and Fig. r1b shows the local density variations near voids. It is clear that different void sizes lead to different magnitudes of local density increase, but all the density profiles near nanovoids in NiCoCr exhibit a characteristic λ -shape. Note void 2 is not shown in Fig. r1b because this void is intercepted by the APT specimen surface, so its local density variation is altered.

As the reviewer mentioned, some voids do not look like downward pointing triangle in Fig. 1, which is probably because of the viewing angle and the smaller size of these voids, so some details are not clearly shown in Fig. 1. For the density profiles shown in the manuscript, we have analyzed multiple voids to confirm the trend shown in these profiles are representative.

Fig. r1 (a) APT reconstruction showing voids in NiCoCr (same as Fig. 1g in the manuscript). (b) Density profiles passing through each void as shown in (a). The reduced density is the local atomic density divided by the average atomic density in the matrix. For the x-axis, lower distance value indicates the region is evaporated earlier during the APT experiment, and the position where local density starts to increase is defined as zero.

4 - How sensitive are the characteristic density profile shapes to reconstruction parameters/models?

Response: CAMECA's IVAS software was used for the APT reconstructions. As discussed in reference ¹⁰, there are five reconstruction parameters: detector efficiency (DF), image compression factor (ICF), initial tip radius R_0 , shank angle and sphere-cone radius ratio. We use

the “tip profile” mode, which has provided us a more accurate reconstruction than the other two modes (i.e., voltage and shank) according to our experience. In this mode, the shank angle and R_0 are determined based on the tip image acquired using a STEM or SEM image before the APT experiment. The DF is dependent on the APT hardware (set to 36%), the ICF is dependent on APT hardware and the specimen geometry (set to 1.65), and the sphere-cone-radius ratio was set to have tangential continuity. These were the best values to obtain an optimized reconstruction based on our correlative experiments. The bubble sizes as calculated from the APT dataset were similar to those measured by STEM. If the reconstruction parameters were set wrong, the bubble sizes and shapes would either be compressed or elongated in the z-direction.

Additional reconstructions have been conducted to test the sensitivity of the density profiles to these parameters. It is found that the DF, ICF, and sphere-to-cone ratio can affect the reconstruction. As an example, in Fig. r2 below, we vary the DF from 0.2 to 0.5 and plot the characteristic density profiles. As discussed previously, the changes in DF can compress or elongate the voids in the z-direction of the reconstruction¹⁰, so the magnitude and width of the local density variations are modified with different DF values, but the characteristic λ - or ω -shape remain unchanged. For APT reconstructions using the other two modes, i.e., voltage and shank, we would expect a similar phenomenon as the effects of DF on the reconstruction should be the same. In the revised manuscript, we added the detailed reconstruction procedures to the Method section.

Fig. r2 Local density variations near nanovoids in APT reconstructions with different detector efficiency (df) values. (a) shows characteristic λ -shape for the low-field case and (b) shows characteristic ω -shape for the high-field case.

5 - Why does the void not appear to have a spherical bottom in Figure 3d?

Response: The non-spherical bottom of the void in Fig. 3d (now Fig. S4d) is a display artefact. In order to represent the void buried in the material matrix, we cut a slice intersecting the void and the slice is about 2 nm thick. Since the void is small (radius=1.5nm), we can see the atoms at the border of the slice, which lead to the non-spherical bottom of the void in Fig. 3d (now Fig. S4d). In the revised manuscript, Fig. 3 is moved to Supplementary Information Fig. S4 because a better

representation of the tip surface evolution is provided in Fig. 5. Please refer to our response to Comment 9 for more details.

6 - The simulations were performed at a smaller length scale than the experimental voids. Do you anticipate any particularly scaling relationships to hold such that you can extrapolate from one size scale to another? To play devil's advocate, one might argue that the volumes scale as r cubed and the shell surface area scales as r squared so wouldn't that potentially affect the evaporation behavior (making it harder to extrapolate the simulations)?

Response: The simulation results at a smaller length scale can be proportionally scaled up, and the simulation is a good comparison to the experiment at a larger length scale. The reviewer's argument on the volume-surface relation is definitely correct, but the ion trajectory is only dependent on the shape of the electrode (i.e., the APT tip) in a dimensionless geometry. One demonstrated using a cylindrical symmetry example was provided in the reference¹¹. Therefore, the ion trajectory will always have the same shape whatever the length scale of the void, as long as the aspect ratio of tip/void radius is unchanged. In our study, we set the tip/void ratio in the simulation to the same value as shown in the experiment. We do find from the simulations that the aberration degree is affected by the tip/void radius ratio, but we plan on publishing this effect in a separate paper.

7- Where did the 30% higher/lower evaporation field numbers come from? How sensitive are the results to these numbers? Some justification should be given for the choice.

Response: We have performed additional simulations to understand how sensitive the simulated results to different shell evaporation fields (E_{shell}).

As shown Fig. r3, we varied E_{shell} from 0.2 to $2E_{\text{matrix}}$ (the evaporation field of the matrix) and obtained the reduced density profile across the nanovoid. Here the "distance from the interface" is the distance from the upper interface between the void segregation shell and the matrix, and smaller distance means the region is evaporated earlier during the APT experiment; the reduced density is the local atomic density divided by the matrix material density. As shown in Fig. r3a, the shape of density variations gradually evolve from a λ -shape to ω -shape as E_{shell} increases from 0.2 to $2E_{\text{matrix}}$. More quantitatively, as shown in Fig. r3b when $E_{\text{shell}} < 0.8E_{\text{matrix}}$, the density variations show a characteristic λ -shape, when $E_{\text{shell}} > 1.1E_{\text{matrix}}$ the density variations show a characteristic ω -shape. In our simulation, $E_{\text{shell}} = 0.7E_{\text{matrix}}$ for the low-field case and $1.3E_{\text{matrix}}$ for the high-field case were chosen because these values generate the best match to the experimentally measured density variations.

A brief discussion and related figures have been added to the Supplementary Information.

Fig. r3 (a) Surface plot showing local density variations near a nanovoid as a function of E_{shell} and the distance from the void/shell interface. (b) Colormap showing local density values near a nanovoid with different E_{shell} . Additional labels are provided in (b) to mark the range of E_{shell} in which a characteristic λ -or ω -shaped density variations can be observed.

8 - Line 242 you state that the “cases prove that the simulations quantitatively reproduced the experiments”. I strongly disagree. The results are not quantitative. If they were quantitative then you should plot them on the same axes. But the y-scale of the data and simulation are quite different, and the x-scale of the data and simulation are quite different. The results match up qualitatively quite well, but in no way do they appear to match up quantitatively. Please remove such language.

Response: Following the reviewer’s suggestions, we have modified the statement as “the *good match in the characteristic density variation shapes* between experiment and simulation ... demonstrates that the evaporation fields of the shell control the local density variations patterns...”. We stress that the agreement here is referring to the characteristic shape of the local density variations, not the magnitude.

In the revised manuscript, we also made the y-scale the same for the experimental and simulation 1D density profiles (Figs. 2a, 2b, 3d and 3f). In the revised figures, the y-axis shows reduced density instead of the absolute atomic density values. The reduced density is defined as the local atomic density divided by the average atomic density in the matrix. The reduced density provides a more general representation of the local density variations introduced by nanovoids as the reduced density is independent from the matrix atomic density, which might be affected by different sample materials and APT running conditions.

9 - Figure 6 is nice in that it gives a simple interpretation for the results of the simulation. However, the reader has not had a chance to see the results of the simulation directly. Is it possible to show some cross sectional (x,z view) snapshots from the simulation at various points?

Basically just the (x,z) view of the data shown in Figure S6. You could leave it in the supplementary, but it would be nice to see the data that the schematic in Figure 6 is derived from.

Response: We have added the cross-sectional snapshots of the simulated tip at different evaporation stages in Fig. 5 in the revised manuscript. We completely agree with the reviewer that these snapshots offer a direct view of the tip morphology evolutions and can greatly facilitate the understandings of the physical mechanisms shown by the schematic plots in Fig. 6. Comparing Fig. 5 and Fig. 6, it is clear that key features of the tip morphology evolutions are accurately represented in the schematic plots.

Since the new Fig. 5 provides a better representation of the tip surface evolutions from simulation than the previous Fig. 3, we moved the previous Fig. 3 to supplementary information Fig. S4.

10 - Line 376 you again state that the simulation “quantitatively reproduced” the data. I again strongly disagree that the agreement is quantitative. It appears qualitative to me. Please remove such language. Perhaps you could plot the “ground truth” compositions that you built into the simulation on Figure 7b to highlight that the shell compositions are not strongly altered despite the trajectory aberrations. That may make your point in a more convincing manner.

Response: Following the reviewer’s suggestions, we have modified the statement as: “*Very similar concentration profiles* can be obtained using the simulated reconstruction (Fig. 7b), assuming a segregation shell surrounding the void with assigned concentration values listed in Table 1. The *qualitative agreement* between the experimental and simulated concentration profiles, *as well as the agreement between the simulated and assigned concentration values*, indicate that...”. We also plotted the ground truth composition as horizontal dashed lines on Fig. 7b as the reviewer suggested.

11 - The data broadly convince me that for a selected evaporation field, shell/matrix composition, and shell/void size, that the concentration observed at the bottom of the void is close to the “real” value, but do you have any idea how much this agreement may change by changing these parameters? Right the claims you make about voids are implied to be relatively broad when only a very small region of parameter space has been explored in simulations and these parameters are not entirely the same as for the experiment. I’d like to see much more emphasis on the qualitative nature of the comparison for your material system – which is quite good and compelling and adds to the knowledge based of the atom probe field – and less on the ‘quantitative’ nature (which is lacking in my view).

Response: We appreciate the reviewer’s insightful comments. As shown in Fig.8 a, the fraction of atoms originating from the shell gets highest near the bottom of the void, so the most accurate concentration measurement can be obtained there. A similar plot is drawn based on our new simulations with different E_{shell} values. As shown in Fig. r4, for E_{shell} from $0.5 E_{\text{matrix}}$ to $2E_{\text{matrix}}$, the fraction of shell atoms either remains 100% or gets higher as it approaches the void bottom, indicating the most accurate segregation measurement can be obtained near the void bottom. As mentioned in the response to the first comment, our ongoing research finds that the magnitude of the ion trajectory aberrations varies as the tip radius/void radius ratio varies, but a trend similar to Fig.8 can still be found.

We completely agree with the comment that more emphasis should be put on the qualitative nature of this study since it is impossible to explore all the parameter space within one work. Therefore, we have added a brief discussion to the second to last paragraph to highlight that the current work focuses on elaborating the physical mechanism of the nanovoid field evaporation process and promoting the APT method for accurate void chemistry characterization. Future investigations are still necessary to fully quantify the effects of different void sizes, shapes, and other parameters on the trajectory aberrations.

Fig. r4 Fraction of shell atoms along the concentration profiles across the void with different shell evaporation field values. The void position is marked by the vertical dashed lines, and the distance zero indicates the location of the upper boundary between the segregation shell and the matrix.

Supporting comments:

12 - Figure S1 is difficult to interpret. The voids in the HAADF images from Figure 1 appear roughly spherical, but the chromium deficient regions of the STEM-EDS image do not appear spherical. A discussion of this discrepancy would be useful here. Perhaps including the HAADF image for the same region as Figure S1a would help.

Response: The HAADF image and a brief discussion have been added to the revised Supplementary Information. In the HAADF image, voids show as dark spheres. Combining the HAADF images with EDS mapping, we can find that several voids locate close to each other, so the Cr concentration gradient introduced by each void interacts with each other. In addition, the EDS measurement projects a 3D sample onto a 2D plane, which can cause neighboring concentration profiles to overlap. Both factors can lead to the non-spherical Cr concentration distribution shown in the EDS mapping.

13 - Why do the profiles in Figure S4 look so different from those in Figure 7a? Especially the cobalt – the shape looks qualitatively different.

Response: The concentration profile in Fig.7a is obtained near voids in NiCoCr while the profile in Fig.S4 (now Fig. S6) is from voids in NiCoCrFe doped with 1 at. % Si and Al. Previous studies have shown that Ni₃(Al, Si) precipitates are commonly observed in Ni-based alloys^{12,13}. It is likely that a similar precipitation process takes place near the void in the doped NiCoCrFe and expels the Co. Meanwhile, Co prefers to segregate to voids as interstitials because of its smaller atomic size¹⁴. These two kinetic processes may compete with each other and lead to the unique shape of Co concentration profiles in Fig. S4 (now Fig. S6).

A thorough explanation of this interesting segregation profile is beyond the scope of this work, and we are conducting another project to understand the segregation behaviors in doped SP-CSAs now. Nevertheless, Fig. S4 (now Fig. S6) is a good example showing the power of the APT method for void chemistry characterization. The concentrations of Si and Al are quite low. It will be very challenging to get segregation profiles with similar accuracy if STEM-based EDS/EELS methods are used.

14 - Figure S5. I think I understand the point you are trying to make here, but not enough detail is given for a real understanding of what the inset is trying to show. Is the appearance of the 'ring-shaped illumination model intensity plot dependent on the ratio of the gaussian width parameter relative to the circle diameter?

Response: We use the inset to illustrate the physical origin of the increased atomic density inside the empty void. We can regard an open void as a ring-shaped illumination device. Although there is no illuminator (i.e., no light source) in the center of the ring, the light intensity in the center is still the highest because of the overlapping light from all the illuminators on the ring. Here the analogy is completely qualitative. The appearance of the intensity plots depends on the detailed values of the gaussian width and the circle diameter as the reviewer pointed out.

Calculation details about the inset of Fig. S5 (now Fig. S8) have been added. We also make it clear to the readers that the inset intensity profile just provides a qualitative explanation to the experimental intensity profile.

15 - The word 'radical' was used in many places instead of 'radial'

Response: All typos have been corrected.

16 - A circular colormap might make Figure S6a/c a easier to interpret.

Response: The circular colormap has been added base on the reviewer's suggestion.

Reviewer #2 (Remarks to the Author):

This paper reports on the interpretation of nanovoids in APT reconstruction using a correlative approach, which is an important issue for several applications. Overall, I have nothing to criticize about the work, which makes it difficult for me to write a constructive review that could help to further improve the manuscript.

Response: We appreciate the reviewer's support of our work and the following comments to better improve the manuscript.

I only have a few minor points:

1- When you used voltage mode to perform the experiments, you didn't detect any He⁺ species? Even though the He pressure corresponds to a few GPa, some He could be ionized. But I guess the background noise of the mass spectra doesn't allow to observe it.

Response: We carefully analyzed the mass spectrum and didn't observe any He⁺ peaks. It is likely that almost all the He atoms escape the pressurized cavities as soon as the bubbles rupture so the He cannot be ionized. As the reviewer suggested, although a very small fraction of He can get ionized, the signal is still too low to be detected compared to the background noise of the mass spectra.

2- You claimed that you didn't find any experimental counterpart for the equal-field case (line 240). It is a bit surprising since you analyzed pure Ni (with voids), and I assume that the shell surrounding the void is also made of Ni, but the sigma explanation is reasonable.

Response: It is worth mentioning that even in pure Ni without composition segregation near voids, the atoms near voids (or bubbles) still experience quite a different physical environment from the atoms in the Ni matrix. First, atoms on the void edges have dangling bonds. Second, the high He gas pressure inside bubbles generates large local deformation in the shell atoms surrounding the bubbles. Both factors can contribute to a different evaporation field of Ni atoms in the shell from atoms in the matrix.

3- It would be good to discuss the domain of applicability of your work. The voids are 4 to 12 nm, but is your approach applicable to bigger voids? Especially, I was thinking of nanoporous Au, with pores in the 20-50 nm size range, that have been filled with Co and Fe filling by e-beam induced deposition (Microsc. Microanal. 2015, 21, 557–563), or filled with Cu by electrodeposition (Mater. Charact. 2017, 128, 269–277 / Electrochim. Acta 2018, 283, 611–618), and also nanoporous Si filled with Ni (Ultramicroscopy 2017, 182, 112–117). Is your approach applicable to such systems?

Response: Nanoporous material is a very interesting research topic for APT. Careful considerations should be taken to decide whether our approach can be applied to the nanoporous samples. In our analysis, the nanovoids (or pores) need to be enclosed in the APT tip for the experiment to work, and the tip diameter usually ranges from tens to a hundred nm. If the pore size is small and the pore density is low, we would expect the characteristic density variations discussed in the manuscript to appear near nanopores in the APT reconstruction. However, if most pores are larger than or close to the APT tip size (e.g. 50 nm pores), it will be challenging to prepare APT tips with enough curvature and mechanical strength. In addition, big parts of the tip may be ripped off even before evaporation due to the discontinuous matrix¹⁵. In this case, as the cited paper suggested, filling the pores by deposition to make compact samples will be a good approach for the following APT analysis. After the pores are filled, they are no longer voids, but should be treated as precipitates or phase boundaries, which have been extensively studied^{16,17}.

4- Could you also briefly discuss the applicability in the case of non-perfectly round-shaped voids?

Response: We did simulations on cubic nanovoids and similar trends were observed in these simulations, i.e. the characteristic λ -shaped and ω -shaped density profiles can be found near cubic voids with different evaporation fields. However, to fully quantify the effects of various void parameters (e.g., void shape, void/tip aspect ratio, etc.), future studies are still necessary, and we are carrying out related investigations now. In the revised manuscript, we added a brief discussion to the second to last paragraph to make this point clear to the readers.

5- On line 304, you say that r is estimated using Eq(1). I think it should be Eq(2) instead.

Response: The typo has been corrected.

6- On lines 446 and 449, there was an issue in the conversion of the pdf, and the unit of 500 appears as a square.

Response: The unit of the 500 should be °C. We will make sure the unit appears correctly in the revised manuscript.

7- In the SI, line 45, you wrote LEAP 4009X HR, I think it should be 4000.

Response: The typo has been corrected. Thank you for pointing this out.

Reviewer #3 (Remarks to the Author):

The presented study is an extremely well executed piece of work that goes to great lengths to get to understand the chemistry of nanovoids in materials as well as possible with currently available methods. I highly recommend the publication of this article. Especially, this is one of the first uses of full 3d field evaporation simulation to aide the understanding of a real word problem in APT. I nevertheless have a few comments about necessary improvements to the manuscript.

Response: We appreciate the reviewer's support of this work and appreciate the time to provide helpful comments to improve the manuscript.

1. Most importantly, since so much effort has gone into this work, it is incredibly dense to read. This is no problem for the few fully initiated into the topic, but for the general APT community it would be very helpful to illustrate the expected image artefacts and how they would impact a trajectory (Bas type) reconstruction in the introduction of the paper. This happens somewhere half way through, where you would already have lost many readers.

Response: Following the reviewer's suggestion, in the third paragraph of the manuscript, we added a very brief introduction to the expected aberration near nanovoids and its effects on the reconstruction. We also added a short introduction to the working principles of APT in the Method section for readers who may not be familiar with APT.

2. In the simulations, a deviation in the evaporation field of 30% with respect to the matrix is assumed. How is this justified? for many d-block metals, the (pure elemental) evaporation fields differ by much less than that. Since much of the shape of the resulting concentration profiles is

dependent on when a "crossover point" of the ion trajectories is reached, surely there would be a development of the observed concentration profiles dependent on the differences in evaporation field.

Response: We did additional simulations to understand the effects of different shell evaporation field (E_{shell}) on the characteristic density profiles.

As shown in Fig. r3, we varied E_{shell} from 0.2 to $2E_{\text{matrix}}$ (the evaporation field of the matrix) and obtained the reduced density profile across the nanovoid. Here the "distance from the interface" is the distance from the upper interface between the void segregation shell and the matrix, and the direction of the distance follows the direction of the APT evaporation process; the "reduced density" is the local atomic density divided by the matrix material density. As shown in Fig. r3a and suggested by the reviewer, the shape of density variations gradually evolve from the λ -shape to ω -shape as E_{shell} increases from 0.2 to $2E_{\text{matrix}}$. When $E_{\text{shell}} < 0.8E_{\text{matrix}}$, the density variations show a characteristic λ -shape; when $E_{\text{shell}} > 1.1E_{\text{matrix}}$ the density variations show a characteristic ω -shape (Fig. r3b). In our simulation, we chose $E_{\text{shell}} = 0.7E_{\text{matrix}}$ for the low-field case and $1.3E_{\text{matrix}}$ for the high-field case because these values generate the best match to the experimentally measured density variations.

We hope to mention a few reasons that may lead to the 30% difference in evaporation fields between the void shell and the matrix. First, atoms on the void edges have dangling bonds. Second, the high He gas pressure inside bubbles generates high local deformation near the bubble. Third, for voids in alloys, the elemental segregation near voids generates a quite different chemical environment for the shell atoms from the matrix.

A brief discussion and related figures have been added in Supplementary Information note 5 to discuss the effects of different E_{shell} on the characteristic density profiles.

Fig. r3 (a) Surface plot showing local density variations near a nanovoid as a function of E_{shell} and the distance from the void/shell interface. (b) Colormap showing local density values near a

nanovoid with different E_{shell} . Additional labels are provided in (b) to mark the range of E_{shell} in which a characteristic λ - or ω -shaped density variations can be observed.

3. Lastly, I am a bit disappointed that figure 6 only contains schematic drawings, since from the simulations, the field evaporation forms at any step during the field evaporation of the nanovoid are available. Pls consider putting cross sections of the nano-voids into the figure, maybe together with the schematics in a left-right half/half, or entirely.

Nevertheless, this is great work and absolutely deserves publishing.

Response: We appreciate the helpful suggestion. The cross-sectional snapshots of the simulated tip at different evaporation stages have been added as a new Fig. 5. We have kept the schematic drawings in Fig. 6 because important quantities (e.g., various radii) can be clearly defined and the physical processes of the void field evaporation can be apparently illustrated using these schematic drawings. However, we completely agree with the reviewer that the snapshots from the simulations offer a direct view of the tip morphology evolutions and greatly facilitate the understandings of the physical mechanisms shown by the schematic drawings. Comparing Fig. 5 and Fig. 6, it is clear that key features of the tip morphology evolutions are accurately represented in the schematic plots. Since the new Fig. 5 provides a better representation of the tip surface evolutions from the simulation than Fig. 3, we also moved the previous Fig. 3 to Supplementary Information Fig. S4.

We again appreciate all the helpful suggestions and comments from the reviewers.

Reference

1. Ding, Q. *et al.* Tuning element distribution, structure and properties by composition in high-entropy alloys. *Nature* **574**, 223–227 (2019).
2. Gludovatz, B. *et al.* Exceptional damage-tolerance of a medium-entropy alloy CrCoNi at cryogenic temperatures. *Nat. Commun.* **7**, 1–8 (2016).
3. Zhang, Y. *et al.* Influence of chemical disorder on energy dissipation and defect evolution in concentrated solid solution alloys. *Nat. Commun.* **6**, 8736 (2015).
4. Newbury, D. E. & Ritchie, N. W. M. Comprehensive Quantitative Elemental Microanalysis with Electron-Excited Energy Dispersive X-ray Spectrometry (EDS): 50 Years Young and Getting Better Every Day! *Microsc. Microanal.* **24**, 712–713 (2018).
5. Kelly, T. F. *et al.* Atom Probe Tomography of Electronic Materials. *Annu. Rev. Mater. Res.* **37**, 681–727 (2007).
6. Egerton, R. F. Electron energy-loss spectroscopy in the TEM. *Reports Prog. Phys.* **72**, (2009).
7. Berta, Y., Ma, C. & Wang, Z. L. Measuring the aspect ratios of ZnO nanobelts. *Micron* **33**, 687–691 (2002).
8. Allen, S. M. & Hall, E. L. Foil thickness measurements from convergent-beam diffraction patterns An experimental assessment of errors. *Philos. Mag. A Phys. Condens. Matter*,

- Struct. Defects Mech. Prop.* **46**, 243–253 (1982).
9. Guo, W. *et al.* Correlative Energy-Dispersive X-Ray Spectroscopic Tomography and Atom Probe Tomography of the Phase Separation in an Alnico 8 Alloy. *Microsc. Microanal.* **22**, 1251–1260 (2016).
 10. Larson, D. J., Prosa, T. J., Ulfing, R. M., Geiser, B. P. & Kelly, T. F. *Local Electrode Atom Probe Tomography*. (Springer New York, 2013). doi:10.1007/978-1-4614-8721-0
 11. Smith, R. & Walls, J. M. Ion trajectories in the field-ion microscope. *J. Phys. D. Appl. Phys.* **11**, 409–419 (1978).
 12. Rastogi, P. K. & Ardell, A. J. The coarsening behavior of the γ' precipitate in nickel-silicon alloys. *Acta Metall.* **19**, 321–330 (1971).
 13. Cho, J. H. & Ardell, A. J. Coarsening of Ni₃Si precipitates at volume fractions from 0.03 to 0.30. *Acta Mater.* **46**, 5907–5916 (1998).
 14. He, M.-R. *et al.* Mechanisms of radiation-induced segregation in CrFeCoNi-based single-phase concentrated solid solution alloys. *Acta Mater.* **126**, 182–193 (2017).
 15. El-Zoka, A. A., Langelier, B., Botton, G. A. & Newman, R. C. Enhanced analysis of nanoporous gold by atom probe tomography. *Mater. Charact.* **128**, 269–277 (2017).
 16. Larson, D. J., Gault, B., Geiser, B. P., De Geuser, F. & Vurpillot, F. Atom probe tomography spatial reconstruction: Status and directions. *Curr. Opin. Solid State Mater. Sci.* **17**, 236–247 (2013).
 17. Larson, D. J., Geiser, B. P., Prosa, T. J. & Kelly, T. F. On the Use of Simulated Field-Evaporated Specimen Apex Shapes in Atom Probe Tomography Data Reconstruction. *Microsc. Microanal.* **18**, 953–963 (2012).

REVIEWERS' COMMENTS:

Reviewer #1 (Remarks to the Author):

I recommend the publication of this article without any further required revisions.

Overall, the authors did a good job addressing the reviewer's comments. My only remaining suggestion (not `required') would be to add Figure r1 (in the response to reviewers) showing the profiles from all the voids to the supplementary material. As a reader I always enjoy seeing how variable a signature is throughout a sample. In any event, the manuscript is great work overall and I enjoyed the opportunity to review it.

Reviewer #2 (Remarks to the Author):

I have carefully read the revised manuscript, the revised SI and the explanatory text, and I can only conclude that the authors have satisfactorily modified the paper with additional content and analysis to answer the main points about which the reviewers previously had reservations and/or questions.

The manuscript will certainly stimulate similar studies by peers in the APT community, and I recommend publication without further revision.

Point-by-point to referees' comment

Dear Referees,

We are grateful to all the helpful comments from the reviewers. The only major change made after the second round of review was the addition of Supplementary Figure 10. Our responses to the reviewers' comments are listed below.

REVIEWERS' COMMENTS:

Reviewer #1 (*Remarks to the Author*):

I recommend the publication of this article without any further required revisions.

Overall, the authors did a good job addressing the reviewer's comments. My only remaining suggestion (not `required') would be to add Figure r1 (in the response to reviewers) showing the profiles from all the voids to the supplementary material. As a reader I always enjoy seeing how variable a signature is throughout a sample. In any event, the manuscript is great work overall and I enjoyed the opportunity to review it.

Response: We appreciate the reviewer's support of our work and the following suggestion to better improve the manuscript. Figure r1 in the response letter has been added to the updated Supplementary Information as Supplementary Figure 10.

Reviewer #2 (*Remarks to the Author*):

I have carefully read the revised manuscript, the revised SI and the explanatory text, and I can only conclude that the authors have satisfactorily modified the paper with additional content and analysis to answer the main points about which the reviewers previously had reservations and/or questions.

The manuscript will certainly stimulate similar studies by peers in the APT community, and I recommend publication without further revision.

Response: We appreciate the reviewer's support of our work